

## The response of sea ice and high salinity shelf water in the Ross Ice Shelf Polynya to cyclonic atmosphere circulations

Xiaoqiao Wang[1], Zhaoru Zhang[1][2*], Michael S. Dinniman[3], Petteri Uotila[4], Xichen Li[5], Meng Zhou[1][2]

[1] School of Oceanography, Shanghai Jiao Tong University, Shanghai, China.

[2] Key Laboratory for Polar Science, Polar Research Institute of China, Ministry of Natural Resources, 200136,Shanghai.

[3] Center for Coastal Physical Oceanography, Old Dominion University, Norfolk, VA 23529, USA.

[4] Institute for Atmospheric and Earth System Research/Physics, Faculty of Science, University of Helsinki, Helsinki, Finland.

[5] International Center for Climate and Environment Sciences, Institute of Atmospheric Physics, Chinese Academy of Sciences, Beijing, China.

Corresponding author: Zhaoru Zhang (zrzhang@sjtu.edu.cn)

**Abstract**

Coastal polynyas in the Ross Sea are important source regions of high salinity shelf water (HSSW) – the precursor of Antarctic Bottom Water that supplies the lower limb of the thermohaline circulation. Here, the response of sea ice production and HSSW formation to synoptic- and meso-scale cyclones were investigated for the Ross Ice Shelf Polynya (RISP) using a coupled ocean-sea ice-ice shelf model targeted on the Ross Sea. When synoptic-scale cyclones prevailed over RISP, sea ice production (SIP) increased rapidly by 20–30% over the entire RISP. During the passage of mesoscale cyclones, SIP increased by about 2 times over the western RISP but decreased over the eastern RISP, resulting respectively from enhancement in the offshore and onshore winds. HSSW formation mainly occurred in the western RISP and was enhanced responding to the SIP increase under both types of cyclones. Promoted HSSW formation could persist for 12–48 hours after the decay of the cyclones. The HSSW export across the Drygalski Trough was negatively correlated with the meridional wind speed, while the export across the Glomar Challenger Trough was positively correlated with the meridional wind. Such correlations are mainly controlled by variations in geostrophic ocean currents that result from sea surface elevation change.



## 1 Introduction

Antarctic coastal polynyas are characterized by the areas of persistent open water surrounded by sea ice along the coastlines, which tend to appear recurrently at fixed geographical locations and periods of the year. These coastal polynyas are mechanically driven by offshore katabatic and synoptic winds (Bromwich et al., 1998; Massom et al., 1998; Morales Maqueda et al., 2004). Katabatic wind is traditionally defined as a downslope cold flow driven by gravity and pressure gradient force over a sloping surface near the Antarctic coast, and its direction is largely controlled by local topography (Lutgens and Tarbuck, 2001). In general, the near-surface wind fields over the Antarctic continent are forced by both katabatic flow and meso- or synoptic-scale atmospheric forcing. New sea ice production within coastal polynyas and the associated brine rejection process leads to the formation of high salinity shelf water (HSSW), which is a precursor of Antarctic Bottom Water (AABW), a key component of the lower cell of the meridional overturning circulation (Comiso and Gordon, 1998; Ohshima et al., 2013; Whitworth et al., 2013). Furthermore, Antarctic coastal polynyas are identified as biological "hot spots" due to the enhanced primary productivity during the austral spring and summer (Arrigo and van Dijken, 2003). The polynya regions are also characterized by massive atmospheric $CO_2$ sinking (Hoppema and Anderson, 2007; Arrigo et al., 2008; Tortell et al., 2012) resulting from deep convection and large amounts of phytoplankton accumulation compared with adjacent waters (Tremblay and Smith, 2007). Consequently, coastal polynya processes play an important role in the global climate system.

Previous studies indicate that the sea ice production rate (SIP) and HSSW formation within the coastal polynyas are significantly increased with the strength of katabatic winds (Mathiot et al., 2010; Barthélemy et al., 2012; Zhang et al., 2015; Dale et al., 2017; Cheng et al., 2019). Most of these studies focused on the role of winds in the Antarctic coastal polynyas on seasonal or longer time scales. However, atmospheric conditions along the Antarctic coast are characterized by high-frequency wind events associated with the passages of synoptic- or mesoscale cyclones (Turner et al., 2009; Chenoli et al., 2015; Weber et al., 2016; Wang et al., 2021). The coastal margin of Antarctica is regarded as the most active cyclogenetic region on earth due to the existence of a strong baroclinic zone around the Antarctic continent (Parish and Cassano, 2003). The coastal Ross Sea, which has been identified as an important source region of AABW due to the presence of the Terra Nova Bay and the Ross Ice Shelf polynyas (Jacobs et al., 1970; Gordon and Comiso, 1988; Whitworth and Orsi, 2006), is frequently affected by the passages of cyclones (Bromwich et al., 1993; Simmonds et al., 2003; Knuth et al., 2011; Uotila et al., 2011; Yu et al., 2019). Recent studies have begun to focus on the influence of synoptic-scale wind forcing on sea ice properties in the Ross Sea polynyas based on the observations. The sea ice properties and the polynya extent have a strong correlation with wind speed in the Ross Sea polynyas (Dale et al., 2017; Cheng et al., 2019; Ding et al., 2020). Thompson et al. (2020) demonstrated using in-situ observations, available from the Polynyas and Ice Production and seasonal Evolution in the Ross Sea (PIPERS) program which conducted an autumn ship campaign in 2017 and two spring airborne campaigns in 2016 and 2017, that the estimated frazil ice production could increase up to 110 cm d$^{-1}$ during the strongest wind events (Ackley et al., 2020). Wenta and Cassano (2020) found that during an extreme wind event associated with the passage of two cyclones, the extent of the Terra Nova Bay polynya (TNB) increased dramatically by over 20-fold. These studies provide important insights into the response of polynyas to changes in atmospheric forcing. However, the influence of cyclones on the oceanic processes, including the convection and the formation of HSSW that directly affect the AABW and thermohaline circulation, has not been revealed. Moreover, different types and paths of cyclones may induce different coastal wind patterns over the polynyas, and result in distinct responses by sea ice production and the HSSW formation. The extent of these processes remains uncertain but would be important for understanding the short-scale variability of HSSW and even the AABW formation.





As very few in-situ measurements have been conducted at the Antarctic coastal polynyas during the
freezing season, numerical models have become indispensable methods to investigate the response of sea
ice and oceanic processes to harsh weather conditions, such as cyclones. In this study, a 5-km resolution
regional ocean-sea ice-ice shelf model for the Ross Sea was employed to investigate the role of meso- and
synoptic-scale cyclones in sea ice production and the HSSW formation in the Ross Ice Shelf polynya (RISP),
which is the largest coastal polynya with the highest SIP over the Southern Ocean (Tamura et al., 2008;
Kern et al., 2009). This manuscript is organized as follows. In Section 2, descriptions of the numerical
model, observational data, and model validation and analysis methods are provided. In Section 3, the
impacts of different types of cyclones on the variations of sea ice production and water mass formation are
presented and interpreted. Discussions on the HSSW exports in the troughs that are major conduits for
HSSW outflow and their relationship with the meridional winds are given in section 4. Section 5 provides
the summary and conclusions.

**2 Date and Methods**
2.1 Model data description
This study utilizes the Ross Sea circulation model as described in Dinniman et al. (2018), which is
implemented with the Regional Ocean Modeling System (ROMS). ROMS combines a primitive-equation,
finite-volume ocean model with a dynamic sea ice model (Budgell, 2005) based on an elastic-viscous-
plastic (EVP) rheology solver (Hunke and Dukowicz, 1997; Hunke, 2001). The sea ice model applies the
two-layer ice thermodynamics following Mellor and Kantha (1989) and Häkkinen and Mellor (1992),
which has been verified that it can well simulate sea ice variables over coastal regions around the Antarctic
including the Ross Sea (Stern et al., 2013; Dinniman et al., 2011, 2015). The ice shelves used in this model
are static, which means that the motion or mass change of the ice sheet, including iceberg calving, is ignored.
The thermodynamic and mechanical effects of the Ross Ice Shelf (RIS) cavity on the adjacent water beneath
are parameterized (Holland and Jenkins, 1999; Dinniman et al., 2011). The momentum, heat, and
freshwater (imposed as a salt flux) fluxes in the open ocean are calculated from the COARE version 3.0
bulk flux formulae (Fairall et al., 2003). There is no relaxation for surface temperature or salinity towards
prescribed values, such as an observational climatology.

The southern extent of the Ross Sea model domain includes most of the cavity under the RIS, and the
northernmost part consists of the continental shelf break and extends to 67.5°S (Fig. 1). The model has a
horizontal resolution of 5 km, and 24 vertical layers with variable thicknesses (higher resolution towards
the top and bottom surfaces) based on a terrain-following vertical coordinate system (Haidvogel et al., 2008;
Shchepetkin and McWilliams, 2009). The topographic datasets used in this model are from the International
Bathymetric Chart of the Southern Ocean (IBCSO) and Bedmap2 (Arndt et al., 2013; Fretwell et al., 2013)
which include the elevation of the bedrock and the base of any floating ice shelves (mainly for the RIS).
The lateral open boundary conditions for temperature and salinity are derived from the climatological data
based on the World Ocean Atlas 2001 (WOA01), and for barotropic velocities from the Ocean Circulation
and Climate Advanced Modelling project (OCCAM; Saunders et al., 1999). Monthly observed data from
passive microwave satellite observations (SSM/I) over 1999–2014 are used as the lateral open boundaries
for sea ice concentration. Ocean tidal currents and the inverse barometer effect are not included. Monthly
climatological precipitation used in this model is derived from the Antarctic Mesoscale Prediction System
(AMPS), a high-resolution atmospheric model over the Antarctic (Powers et al., 2003; Bromwich et al.,
2005). Monthly cloud fraction climatology data comes from the International Satellite Cloud Climatology
Project stage D2 (ISCCP D2; Rossow et al., 1996), which is used to calculate net longwave and shortwave
radiations following Berliand (1952). Other atmospheric forcing fields including 6-hourly winds and air
temperature and monthly sea level pressure and humidity, were obtained from the ERA-Interim reanalysis



product (Dee et al., 2011) produced by the European Centre for Medium-Range Weather Forecasts
(ECMWF) (Dinniman et al., 2018). These atmospheric variables are used in the bulk formulae to generate
the surface fluxes in the Ross Sea model (Fairall et al., 2003). The ERA-Interim has a spectral T255
horizontal resolution which corresponds to approximately 79 km spacing on a reduced Gaussian grid. The
ERA-Interim reanalysis products can well resolve the meso- and synoptic-scale cyclones (Uotila et al.,
2013; Chenoli et al., 2015; Yu et al., 2019). The Ross Sea model simulation spans from 15 September 1999
to 15 September 2014 after a 6-yr spin-up simulation, and the model results are output as 5-day-average
values. In this study, for the selected periods of synoptic-scale or mesoscale atmospheric events, model
simulations over June to September of 2005 and 2014 are output as 6-hourly results, which is essential to
revealing the detailed processes over the short duration of the cyclone events.

2.2 Observational data for model validation
In this work, the simulated sea ice concentration in the RISP was validated by the daily Bootstrap Sea Ice
Concentrations from Nimbus-7 SMMR and DMSP SSM/I-SSMIS archived at the National Snow and Ice
Data Center (NSIDC) (Markus and Cavalieri, 2000; Comiso, 2017), which have a horizontal resolution of
25 km (https://nsidc.org/data/nsidc-0079/versions/3). The Advanced Microwave Scanning Radiometer-
Earth Observing System (AMSR-E) product derived SIP was employed to evaluate the modelled SIP. The
AMSR-E SIP is estimated based on the heat flux calculation using a thin-ice-thickness estimation algorithm
and surface atmospheric data (http://wwwod.lowtem.hokudai.ac.jp/polar-seaflux), assuming that the
contribution of oceanic heat flux to sea-ice freezing/melting process is negligible (Nihashi and Ohshima,
2015; Nihashi et al., 2017). The AMSR-E SIP dataset was calculated over 2003–2010, and the annual
cumulative SIP is defined as the integrated ice production from March to October, i.e. the freezing season
(Nihashi and Ohshima, 2015). Wind speeds at 10 m from the ERA-Interim reanalysis were compared with
measured 10-m wind data at the McMurdo Station near the Ross Island (Fig. 1b) over austral winter. The
wind speed data are available at the Reference Antarctic Data for Environmental Research (READER)
project website (http://legacy.bas.ac.uk/met/ READER). We selected two winters to study the influence of
cyclone events, which are featured by high correlations between wind speed from ERA-Interim and
observations (correlation coefficients R>0.5 and p-values P<0.0001), and have representative meso- or
synoptic-scale cyclone events. The selected years were 2005 (R=0.56) and 2014 (R=0.61).

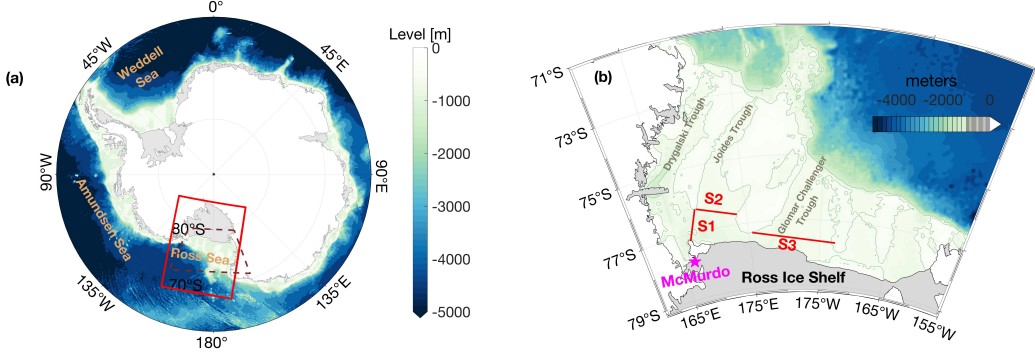


**Figure 1.** Geographic map of (a) the Southern Ocean south of 60°S, and (b) the Ross Sea. Areas in white
show continental surfaces, and areas in light grey indicate the ice shelves. The color scale indicates the
bathymetry. In (a), the Ross Sea model domain is shown by the red solid box, and the brown dashed box
represents the area shown in (b). In (b), the magenta pentacle indicates the position of McMurdo Station,





and the red lines indicate the S1, S2, and S3 sections crossing the troughs that are the major passages for
HSSW outflows towards the slope.

2.3 Model validation
The performance of the Ross Sea model in reproducing sea ice concentration in the RISP was evaluated by
comparing the annual cumulative SIP from the model simulation with the estimates from satellite data. The
annual cumulative SIP averaged over 2003–2010 in the RISP presents similar spatial patterns between the
simulation and observations (Figs. 2a–b). Compared with the observations, the model slightly
overestimates the SIP, which is a common problem with the Southern Ocean ocean-sea-ice models used
for studying Antarctic coastal polynyas (Zhang et al., 2015; Kusahara et al., 2017; Wang et al., 2021). The
integrated SIP volumes over the RISP are $6.74 \cdot 10^2$ and $5.76 \cdot 10^2$ km$^3$ yr$^{-1}$ for modelled and observed datasets
respectively. Such a SIP overestimation is possibly associated with the relatively coarse resolution of the
atmospheric forcing fields compared to the actual wind conditions. The atmospheric model with too low
resolution possibly extends orographic slopes seaward, beyond the actual coastline, and thus induces too
strong offshore winds, which in turn enhance SIP in the coastal polynyas (Stössel et al., 2011). Statistical
analysis of comparisons for daily SIP anomalies in winters (June–September) of 2005 and 2014 from
climatological values was conducted, and the results are shown in Figs. 2c and 2d. Correlation coefficients
between the modeled and observed sea ice concentration in RISP in 2005 and 2014 are 0.46 (P<0.001) and
0.82 (P<0.0001), respectively (Figs. 2c–d), suggesting that the model captures the daily variability of ice
concentration well.

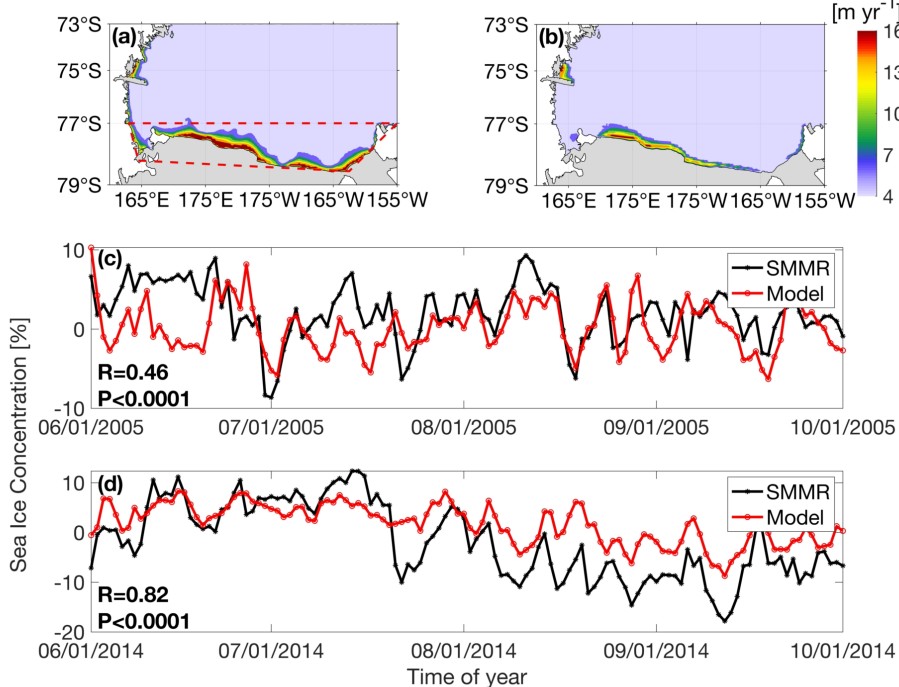


**Figure 2.** (a–b) Annual cumulative sea ice production rates over the Ross Sea from (a) the Ross Sea model
and (b) the AMSR-E product averaged over 2003–2010. The polygon in (a) enclosing the Ross Ice Shelf
Polynya was used to analyse the HSSW characteristics. (c–d) Time series of observed (black lines) and



modelled (red lines) daily polynya-averaged sea ice concentration anomalies in June–September of 2005
and 2014.

2.4 Analysis methods
The extent of RISP was defined as the area where the multi-year-average annual cumulative SIP is greater
than zero based on the Ross Sea model results. The HSSW is defined as the water mass with potential
density ($\sigma_\theta$) above 28 kg m$^{-3}$ (Yoon et al., 2020). Temperature-salinity (T-S) analysis was conducted over
the defined RISP polygon region presented in Fig. 2a to elucidate the impacts of cyclones on the HSSW
formation. The HSSW export rates were calculated across three transects (Fig. 1b) located over three
troughs: the Drygalski Trough (S1), the Joides Trough (S2) and the Glomar Challenger Trough (S3) that
are the major HSSW outflow passages.

Cyclones were tracked by the University of Melbourne Automatic Cyclone Tracking Scheme (Murray and
Simmonds, 1991), based on the ERA-Interim reanalysis product from 1999 to 2014. The optimal
parameters used in this scheme, including the horizontal air pressure field smoothing parameter, the radius
used for the calculation of Laplacian pressure, and the maximum topographic height for detecting the
cyclones, adopted the values by Uotila et al. (2009). The identified cyclone properties included their
locations, lifetimes, and mean radii among others. Cyclones were selected according to the criteria that they
should have lifetimes longer than 12 hours and the distance between their first and last detected locations
should be greater than 1000 km. Such criteria are likely to exclude detected but unrealistic cyclones (Uotila
et al., 2011). We divided the area south of 42°S into 720 sectors, each one spanning 4° in latitude and 6° in
longitude, and calculated cyclone track densities, defined as the number of cyclone tracks per each sector
(Uotila et al., 2013) over 1999 to 2014 (Fig. 3). The cyclones were categorized into two types depending
on their horizontal scale: synoptic-scale cyclones with length scales 1000 km or longer, and mesoscale
cyclones with shorter than 1000 km length scales (Heinemann, 1990; Bromwich, 1991; Carrasco et al.,
2003; Uotila et al., 2011). In this study, we selected two representative synoptic-scale cyclone events that
had different paths over the study region, and one mesoscale cyclone event. These three events occurred in
June 2005 (labeled as MESO), July 2005 (labeled as SYNO1) and September 2014 (labeled as SYNO2).

A three-dimensional momentum analysis of ocean flow was conducted to elucidate the potential
mechanisms for the HSSW export variability under the impact of a typical cyclone. Each momentum term
was analyzed to diagnose the dominant term related to the export variability. The three-dimensional along-
shelf and cross-shelf momentum equations are
$$\frac{\partial u}{\partial t} = -\left(u\frac{\partial u}{\partial x} + v\frac{\partial u}{\partial y} + w\frac{\partial u}{\partial z}\right) + fv - \frac{1}{\rho_0}\frac{\partial p}{\partial x} + K_V\frac{\partial^2 u}{\partial z^2} + K_H\left(\frac{\partial^2 u}{\partial x^2} + \frac{\partial^2 u}{\partial y^2}\right) \tag{1}$$

and
$$\frac{\partial v}{\partial t} = -\left(u\frac{\partial v}{\partial x} + v\frac{\partial v}{\partial y} + w\frac{\partial v}{\partial z}\right) - fu - \frac{1}{\rho_0}\frac{\partial p}{\partial y} + K_V\frac{\partial^2 v}{\partial z^2} + K_H\left(\frac{\partial^2 v}{\partial x^2} + \frac{\partial^2 v}{\partial y^2}\right), \tag{2}$$

respectively, where $u$ and $v$ are the alongshore and cross-shore components of velocity, $f$ is the Coriolis
parameter, $\rho_0$ is the reference density of 1025 kg m$^{-3}$, $p$ is pressure, $x$ is the along-shelf coordinate, $y$ is the
cross-shelf coordinate, $K_V$ and $K_H$ are the vertical and horizontal eddy viscosity coefficients respectively.
Vertical momentum and tracer mixing were calculated using the K-profile parameterization (KPP; Large
et al. 1994). Each term of the momentum equations was calculated for each model time step and output as
6-hourly averages.



## 3 Results and Discussions

### 3.1 Cyclone track densities

The track density of synoptic-scale cyclones can be up to 10–14 times higher than that of mesoscale cyclones in the Ross Sea and the surrounding regions (Fig. 3). Such discrepancy between synoptic-scale and mesoscale cyclones could be related to the relatively coarse spatial resolution of the ERA-Interim product, which may not capture all smaller systems like mesoscale cyclones (Condron et al., 2006; Uotila et al., 2009; Uotila et al., 2011). However, the spatial distributions of cyclones revealed in this study are consistent with the features in Uotila et al. (2011), which are derived from the AMPS high-resolution dataset. The high track density of synoptic-scale cyclones extends to the continental shelf and coastal regions in the western Ross Sea (Fig. 3a). For mesoscale cyclones, a large number of track densities appear at the center of the Ross Sea (at about 180° meridional) in front of the RIS central region (Fig. 3b), which may be related to the cold air outbreaks from the continental interior (Seefeldt and Cassano, 2008; Turner et al., 2009). For synoptic-scale cyclones, two events were selected for our study, which occurred in July 2005 and September 2014, respectively. The first event originated in the area north of the central Ross Sea, and then moved south-eastwards before finally reaching the eastern coastal region. The path of the cyclone related to this event was located in the area with high track densities shown in Fig. 3a. The onset and development of the second cyclone in September 2014 event were primarily situated along the northeastern part of the Ross Sea (about 130°W), accompanied by a slight east-west movement. Although the latter event has a different trajectory than the earlier one, it was also located in the high track-density area (Fig. 3a). For the mesoscale system, we chose one representative event occurring in June 2005. This cyclone moved southeastward from the north-western continental shelf region and lingered in the central Ross Sea, where the track density is higher compared to nearby areas (Fig. 3b).

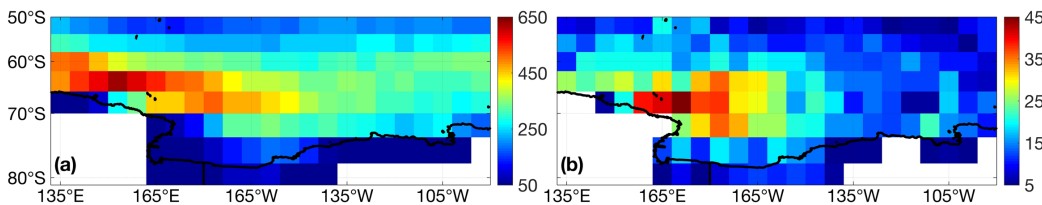

**Figure 3.** (a–b) Accumulated track densities (the number of tracks per section) of (a) synoptic-scale cyclones and (b) mesoscale cyclones in the Ross Sea and surrounding regions over 1999–2014.

### 3.2 Synoptic-scale cyclones

#### 3.2.1 The SYNO1 case

As mentioned in Sections 2.4 and 3.1, two representative synoptic-scale cyclones were selected over the Ross Sea region in the freezing season of 2005 and 2014 respectively. The SYNO1 occurred from July 13 to July 17 of 2005. The center of this cyclone was located northeast of the Ross Sea with a diameter of about 2000 km (Fig. 4). The cyclone developed from 18:00 of July 13 to 06:00 of July 16 (Figs. 4a–f), and in this time period there was a dramatic increase in the offshore wind over the entire RISP, which was associated with the western branch of the cyclone. Corresponding to the wind change, the SIP values increased from 0.1 to 0.3 m day$^{-1}$ (Figs. 5a–f). Following the reduction of wind speed as the cyclone weakened and moved slightly east at 18:00 of July 16 (Fig. 4g), SIP decreased quickly to ~0.2 m day$^{-1}$ over the west side of RISP and ~0.1 m day$^{-1}$ over the east side (Fig. 5g). The coastal winds turned onshore at





06:00 of July 17 when the cyclone weakened (Fig. 4h), and SIP decreased further (Fig. 5h), indicating near-
instantaneous response of sea ice formation to the wind changes. The differences in the meridional wind
speed and SIP between the normal and cyclone conditions averaged over the RISP are summarized in Table
1. For SYNO1, the wind speed was about 1.4 times larger than the normal values, i.e. from 4.8 to 6.5 m s$^{-1}$,
while polynya-averaged SIP increased by 23% reaching 0.043 m day$^{-1}$.

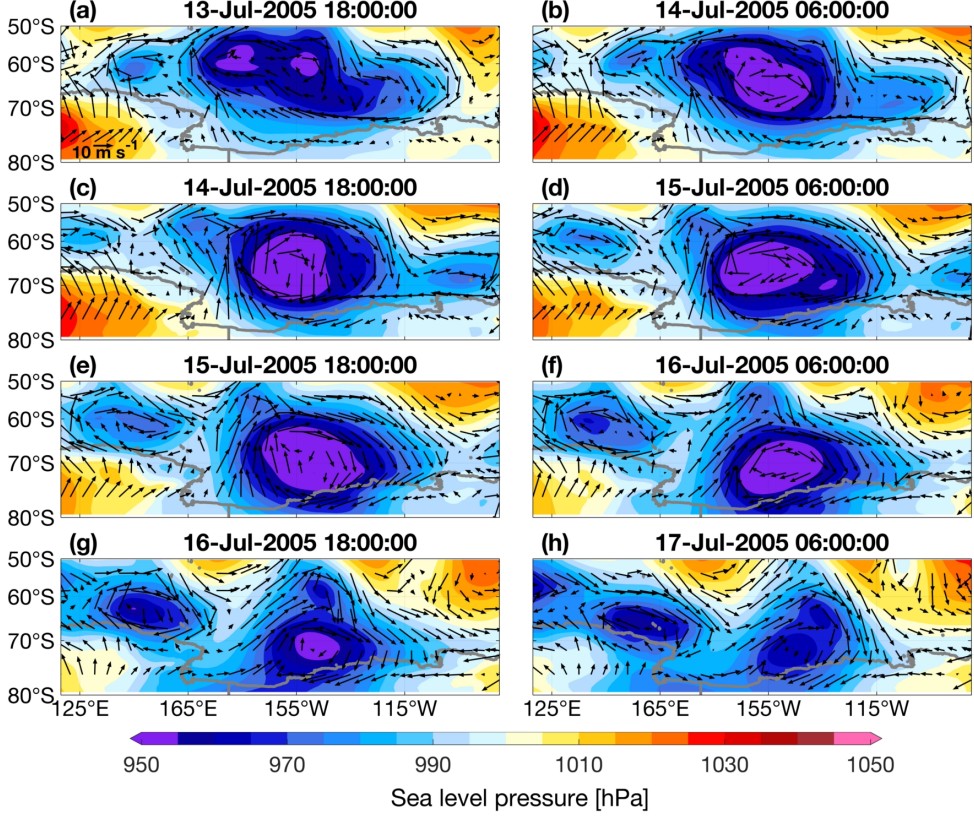


**Figure 4.** (a–h) Spatial distributions of 12-hour-average sea level pressure (color shading) and 10-m wind
vectors (black arrows) in the Ross Sea and surrounding regions over 13–17 July 2005.



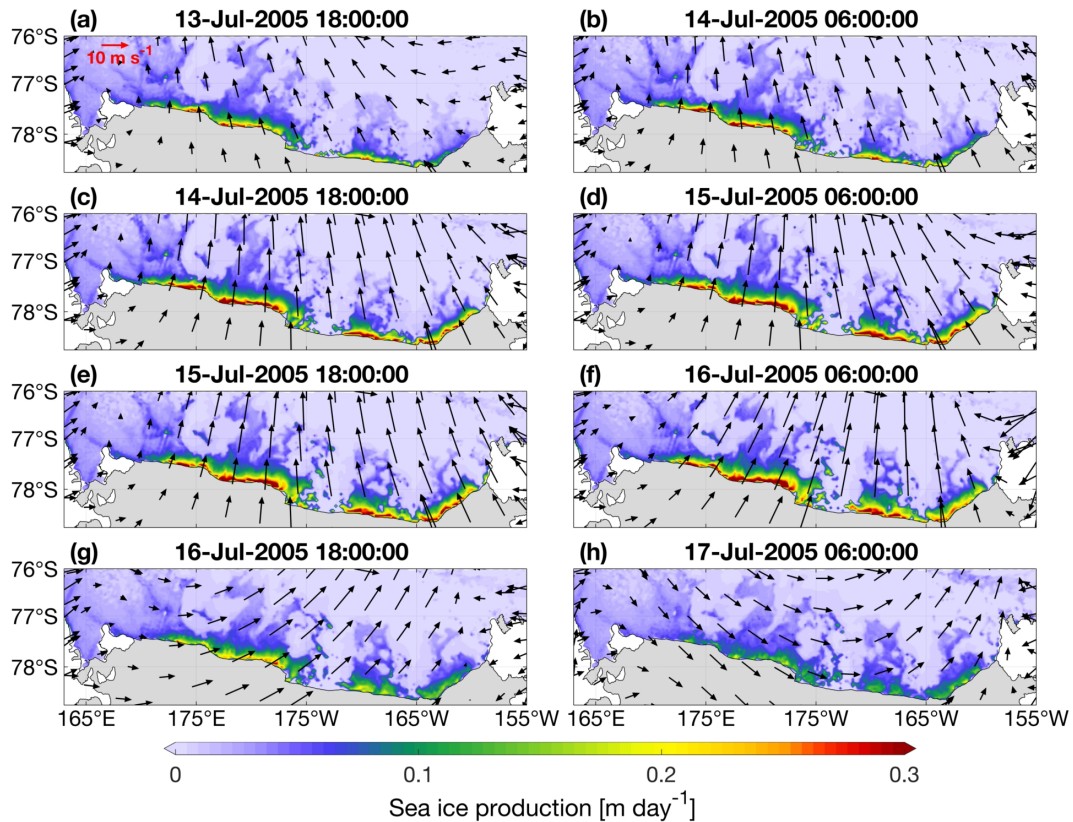

**Figure 5.** (a–h) Spatial distributions of 12-hour-average wind vectors and sea ice production (color shading) in the Ross Ice Shelf Polynya over 13–17 July 2005.

**Table 1.** Comparisons of polynya-averaged sea ice production rates and HSSW properties under the normal and selected synoptic- and meso-scale cyclones (SYNO1, SYNO2 and MESO) for the RISP. The normal values are calculated prior to these events, from 06:00 of July 13 to 18:00 of July 13 for SYNO1, 12:00 of September 16 to 00:00 of September 18 for SYNO2 and 18:00 of June 20 to 12:00 of June 21 for MESO. The cyclone time ranges used in the calculation are consistent with those shown in Figs 4, 8 and 11.

| Properties | | SYNO1 | | SYNO2 | | MESO | |
|---|---|---|---|---|---|---|---|
| | | Normal | Cyclone | Normal | Cyclone | Normal | Cyclone |
| Meridional wind speed (m s$^{-1}$) | | 4.8 | 6.5 | 2.8 | 5.2 | -3.3[a] | 7.1[a] |
| Sea ice production (m day$^{-1}$) | | 0.035 | 0.043 | 0.029 | 0.037 | 0.026[a] | 0.045[a] |
| HSSW volume ($10^4$ km$^3$)[b] | | 3.18 | 3.22 | 3.54 | 3.59 | 2.96 | 2.97 |
| HSSW export (Sv) | S1 | -1.16 | -1.12 | 1.06 | 0.58 | -0.35 | -0.55 |
| | S2 | 0.22 | -0.85 | 0.51 | -0.44 | 0.35 | 0.65 |
| | S3 | 2.01 | 1.05 | 0.79 | 1.25 | -0.06 | 1.63 |

[a]The calculation was conducted for the region west of 175°W within the defined RISP, as the eastern region is dominated by onshore winds, resulting in no significant change in SIP.

[b]The calculation was performed with a 24-hour lag for synoptic-scale events and a 12-hour lag for mesoscale event respectively, based on the discovered lag time of HSSW volume response to the cyclones.






The water mass properties and HSSW volume in the RISP region during the SYNO1 event are illustrated
in Fig. 6. Note that HSSW was mainly formed in the western section of RISP (163°E–175°E) and fresher
water accumulated from 175°E to 155°W (Fig. 6a), which is consistent with previous studies that observed
the highest HSSW accumulation in the western sector of the Ross Sea (Jacobs et al., 1985; Budillion et al.,
2003 ; Mathiot et al., 2012). The amount of HSSW with higher salinity (34.95–35 psu) increased over the
western coastal region in 163°E–164°E from 18:00 of July 14 to 06:00 of July 16 (Figs. 6d–g), when a
dramatic increase in SIP occurred in this area accompanied by the intensification of the SYNO1 cyclone.
The HSSW salinity reached about 35 psu at 06:00 of July 15 (Fig. 6e) when the cyclone had intensified for
2 days. The increase in the higher-salinity HSSW amount persisted for 2 days after the decay of the cyclone
(Figs. 6h–k), and then the amount started decreasing (Figs. 6l–m). The HSSW volume in the RISP area
kept increasing when SYNO1 intensified from 18:00 of July 14 to 06:00 of July 16 and reached the
maximum at 06:00 of July 17 (Fig. 6n), so the HSSW formation could persist at least 1 day after the cyclone
weakened. For the SYNO1 event, the HSSW volume increased by $0.04 \cdot 10^4$ km$^3$ compared to the value
before the cyclone.

The HSSW exports across the three selected transects (S1, S2 and S3 in Fig. 1b) were calculated and related
to the changes in meridional winds (Figs. 7a, b). The northward winds increased slightly from 18:00 of July
13 to 18:00 of July 14 and then turned onshore at 06:00 of July 17 (Figs. 7a, b), associated with the evolution
of the SYNO1 cyclone. The export of HSSW across S1 has a significant negative correlation with the
meridional wind speed (R=-0.70, P=0.012), suggesting that the HSSW had stronger eastward (negative)
transport across the meridionally directed transect S1 when the wind speed increased. The transport across
the zonally directed transect S3 significantly and positively correlated with the meridional wind speed
(R=0.75, P=0.005). The averaged current velocity on S1 and S3 both have strong positive correlations with
HSSW export (R>0.98 and P<0.0001, not shown), suggesting that the velocity is the dominant factor
regulating the export (Figs. 7a, b). Time series of HSSW export and wind speed for Transect S2 are not
shown as no significant correlation between these two variables was detected. The vertical sections of
potential density and circulation along the transects S1–S3 were analyzed to identify physical mechanisms
behind the correlations discussed above (Figs. 7c–n). When the offshore wind speed decreased between 14
and 17 July (Fig. 7a), there was no significant change in the distribution of HSSW along S1 (Figs. 7f, i and
l). Meanwhile, there was notable change in the cross-transect current velocity: the negative (eastward)
velocity in the upper layer between 76.7°S and 76.5°S decreased significantly, while the positive (westward)
velocity in the bottom layer increased (Figs. 7f to i and l). Both features could lead to enhanced exports
when the wind speed decreased. When the offshore wind speed increased between 06:00 and 18:00 of July
13 (Fig. 7a), changes in the cross-transect current velocity on S1 were exactly the opposite of those found
during the decrease of the wind (Figs. 7c to f). For S3, the distribution of HSSW had a slight change, while
the change in velocity is much more pronounced between 178.4°E and 178.7°W, where the positive
(northward) velocity decreased on the west side and the negative (southward) velocity increased on the east
side (Figs. 7h, k and n) when the offshore wind speed decreased (Fig. 7b), resulting in a decrease in HSSW
export which is positively correlated with the wind speed. The dynamical mechanisms for the change in
currents will be discussed in Section 3.4. For S2, although there is no significant correlation between the
HSSW export and the wind speed, the vertical distribution shows a decrease in northward export (Figs. 7d,
g, j and m).


**Figure 6.** (a–m) Temperature–salinity (T–S) diagrams for the RISP region shown in Fig. 2a. The T–S dots
are color-coded with longitude. The black isoline denotes the potential density contour of 28 kg m⁻³. (a) T–
S diagram covering the entire salinity range at 18:00 on July 13, 2005. (b–m) T–S diagrams are bounded
by the purple box shown in (a) and represent values during the SYNO1 event from July 13 to July 19 of
2005. (n) Time series of HSSW volume over the RISP from 18:00 of July 13 to 06:00 of July 19 2005.

339

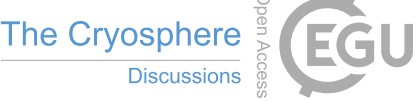

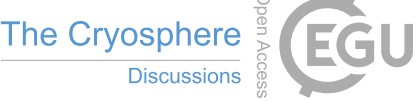

**Figure 7.** (a) Time series of averaged meridional winds along the S1 transect (see Fig. 1), HSSW exports across the S1 and averaged current velocity along the S1 from 06:00 of July 13 to 18:00 of July 18 2005. The gray shading represents the time of the SYNO1 event. The correlation coefficient R and P-value were calculated between the HSSW export and meridional winds. (b) Same as Fig. 7a but for S3. (c–n) Vertical sections of cross-transect current velocity (color shading) and potential density (contour lines) on (c, f, i and l) S1, (d, g, j and m) S2 and (e, h, k and n) S3 at four selected time moments (indicated by the magenta triangles in (a) and (b)). Positive values denote currents from the RISP toward the open ocean, i.e. westward for S1 and northward for S2 and S3. The bold black contour line indicates the isopycnal of 28 kg m$^{-3}$.



### 3.2.2 The SYNO2 case

The second selected synoptic-scale cyclone (SYNO2) developed from 00:00 of September 18 to 12:00 of September 22 2014, and was located on the eastern side of the Ross Sea close to the Amundsen Sea, which is further east than the SYNO1 event (Fig. S1). The entire period of this synoptic-scale cyclone can be divided into three stages. In Stage I, the cyclone developed and the low-pressure center expanded in all directions (Figs. S1a–d). In Stage II, the center moved eastward (Figs. S1e–g). In Stage III, the cyclone rapidly decayed (Figs. S1h–j). The spatial pattern of SIP in the RISP is displayed in Fig. S2. During Stage I, SIP increased over the entire RISP due to the strong offshore winds (Figs. S2a–d), and the increase was more pronounced on the western side of the polynya compared to the eastern side. As the cyclone entered stage II, SIP showed the opposite changes over the eastern and western sides compared to Stage I, when the offshore wind over the western polynya was still strong but the wind over the eastern polynya had significantly turned and weakened (Figs. S2e–g). There was a notable SIP decrease over the entire RISP when the SYNO2 cyclone decayed in Stage III (Figs. S2h–j). Similar to SYNO1, SIP quickly responded to the variation of winds over the entire RISP. For the SYNO2 event, the area-averaged wind speed increased by almost 2 times than before cyclone's arrival, reaching 5.2 m s$^{-1}$. The area-averaged SIP showed an increase of about 28%, reaching 0.037 m day$^{-1}$ (Table 1).

For the water mass response, HSSW volume in the RISP increased significantly until 00:00 on 22 September and then remained at $3.62 \cdot 10^4$ km$^3$ for at least 60 hours (Fig. S3a). The salinity of newly formed HSSW increased to 35.1 psu during Stage I of the SYNO2 cyclone (Figs. S3b–e), but slightly decreased at 00:00 of September 20 (Fig. S3f), possibly associated with the decreased SIP at the eastern side of RISP (Fig. S2e). Afterwards, the formation of higher-salinity HSSW kept increasing near 163°E for 36 hours when the coastal SIP was already decreasing (Figs. S2 and S3k–m). Therefore, both events (SYNO1 and SYNO2) revealed persistent impacts of cyclones on the HSSW formation over the western RSIP, even after these weather systems had decayed. The HSSW exports across the three transects (S1–S3) were calculated during the SYNO2 event but are not shown as the meridional winds and the HSSW exports across S1 and S2 did not correlate. For S3 on the other hand, there was a positive significant correlation between the meridional wind speed and the HSSW export 12-hours later (R=0.51, P=0.055). As shown in Table 1, there was an increase in the HSSW volume during SYNO2, being $0.05 \cdot 10^4$ km$^3$ larger than the rate before the event. The HSSW export across S3 increased about 58% when the wind speed increased from 2.8 to 5.2 m s$^{-1}$, while the export across S1 and S2 decreased.

### 3.3 The MESO case

The selected mesoscale cyclone event was present around RISP from 12:00 of June 21 to 06:00 of June 23, 2005, which is associated with the synoptic cyclone located farther northwest of the Ross Sea (not shown). The formation of this MESO case is consistent with earlier studies, that demonstrate the small synoptic systems could merge into mesoscale cyclones (Carrasco and Bromwich, 1993; Uotila et al., 2009). From the patterns of sea level pressure and wind vectors (Fig. 8), the center of the cyclone was located in the middle of the Ross Sea (Figs. 8c–f), and the horizontal length scale ranged from 500–1000 km. The trajectory of this cyclone was located in the area of high track densities of the Ross Sea (Fig. 3b), suggesting that it is a typical mesoscale cyclone for this region. An enlargement of the wind field is shown in Fig. 9 on top of the SIP field. During the initial stage of MESO (Figs. 9a, b), the entire RISP was influenced by the southern branch of the cyclone and the prevailing alongshore wind. There was little variation in SIP, as the offshore component of wind did not significantly change. As the center of the cyclone moved south and approached RIS (Figs. 8c–e), the western and eastern sections of RISP were respectively affected by the southerly and northerly winds induced by MESO. As a result of the enhanced offshore winds, SIP in the western section increased rapidly to over 0.3 m day$^{-1}$ (Figs. 9c–e). When the cyclone center moved further




south onto the ice shelf and the cyclone winds weakened (Figs. 8f–h), there was a notable decrease of SIP,
suggesting that SIP responds quickly to the winds (Figs. 9f–h). In contrast to the western polynya, SIP in
the eastern polynya presented a slight decrease during MESO (Figs. 9a–e), which was due to the onshore
winds generated by MESO that shrunk the polynya. The response of SIP in RISP was instantaneous during
these selected cyclones, which is consistent with the variation for SIP during typical strong wind events in
East Antarctic coastal polynyas including the Prydz Bay and Shackleton polynyas (Wang et al., 2021).
During MESO, area-averaged SIP increased by 73% than before the cyclone's arrival (Table 1), reaching
0.045 m day$^{-1}$, while the meridional wind speed rose from -3.3 to 7.1 m s$^{-1}$. Thompson et al. (2020) proposed
that the intensity of ice production could rise up to 1.1 m day$^{-1}$ during the windiest events in TNB, which
was calculated from the salt budget using conductivity–temperature–depth (CTD) profiles. Meanwhile, the
maximum SIP rate in the RISP during MESO was 1.2 m day$^{-1}$ in our study, which is just slightly different
from the value in TNB. Differences in topography and location between the RISP and TNB could lead to
such slight differences in atmospheric and hydrological conditions.

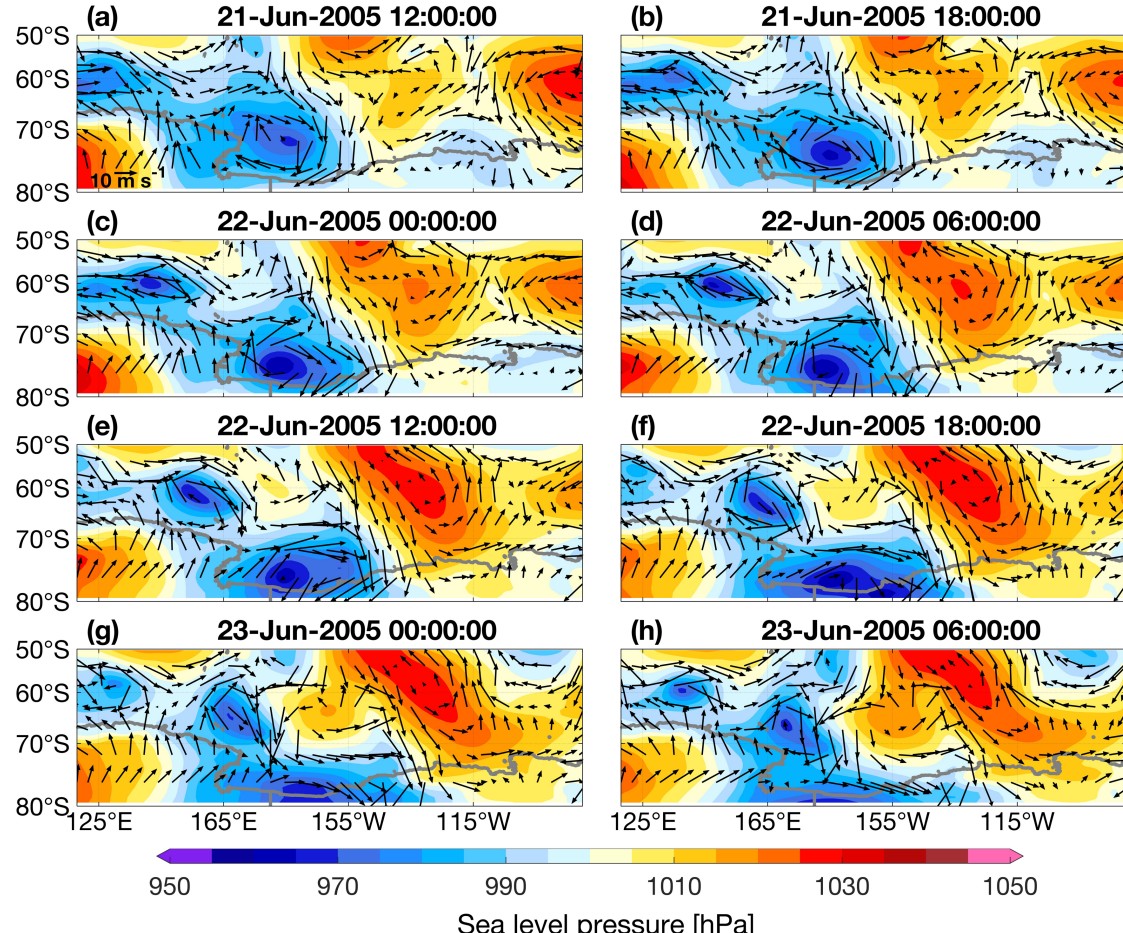

**Figure 8.** (a–h) Spatial distributions of 6-hour-average sea level pressure (color shading) and 10-m wind
vectors (black arrow) in the Ross Sea and surrounding regions over 21–23 June 2005.



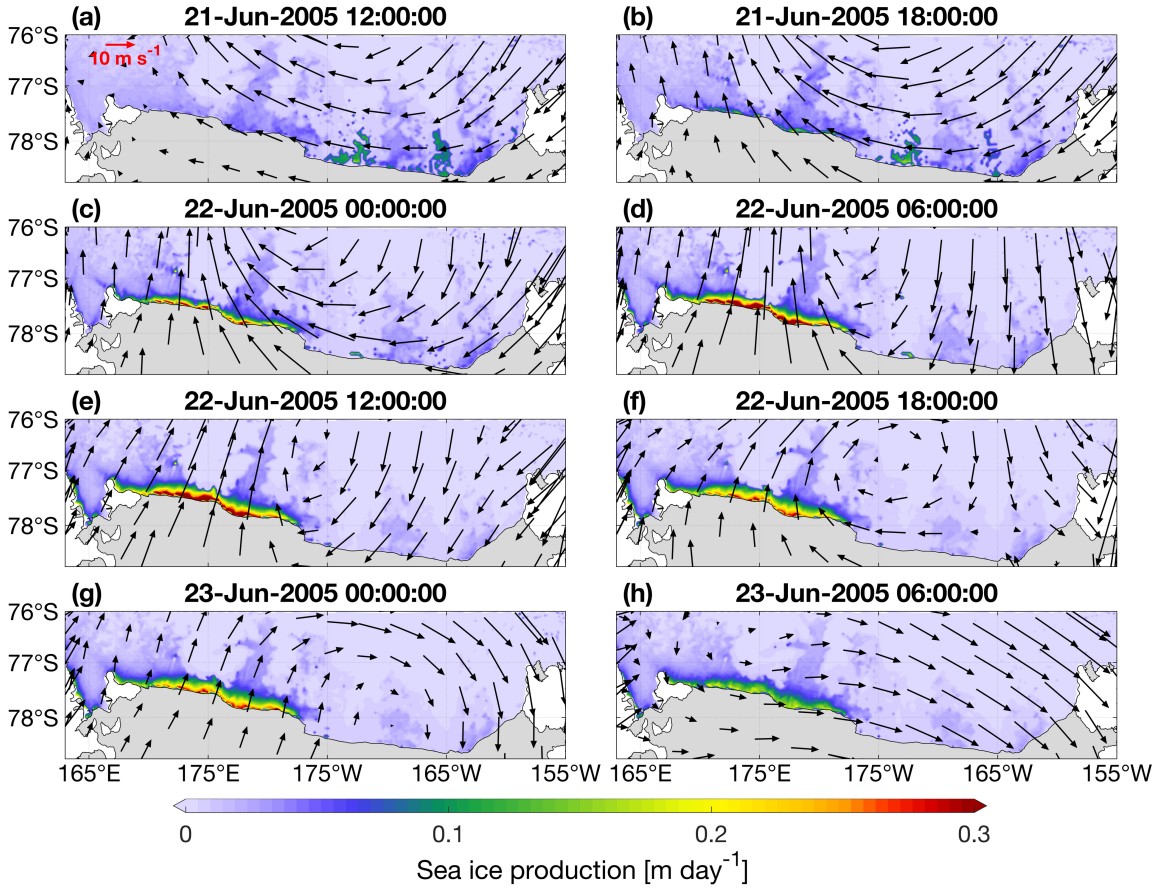

**Figure 9.** (a–h) Spatial distributions of 6-hour-average wind vectors and sea ice production (color shading) near the Ross Ice Shelf Polynya over 21–23 June 2005.

Figure 10a represents a continuous increase for HSSW volume over MESO until 18:00 on June 23 when the cyclone already disappeared. Furthermore, it can be seen from the T-S diagram of Fig. 10 that HSSW with salinity above 34.9 psu began accumulating on early June 22 (Fig. 10e), responding to the increase in SIP (Fig. 9d). The volume and salinity of HSSW increased persistently (especially in the area of 163–166°E) even during the cyclone decay when SIP was already decreasing from late June 22–23 (Figs. 9e–h and 10a, f–k). A notable feature is that the HSSW salinity was continuously in the range of 34.9–34.95 psu from 12:00 to 18:00 of June 23 (Figs. 10j–k), which could be associated with a strong convection or other mixing processes. Afterwards, there was a decrease in the HSSW salinity at 00:00 am of June 24 (Fig. 10l), after about 18 hours of the departure of MESO. These features suggest that the response of the HSSW formation to a mesoscale cyclone could persist for 12–18 hours, which is a comparable time scale to synoptic cyclone events, but with a shorter time lag. The mean HSSW volume during MESO increased slightly by $0.01 \cdot 10^4$ km$^3$ compared to the volume before it. This is much smaller than changes under the synoptic cyclones SYNO1 and SYNO2 (Table 1). This difference is likely related to different spatial scales of cyclones. During MESO, only the western part of RISP was dominated by offshore winds due to the smaller spatial extent of the mesoscale cyclone, which resulted in a smaller region for HSSW production than that under synoptic-scale cyclones during SYNO1 and SYNO2.





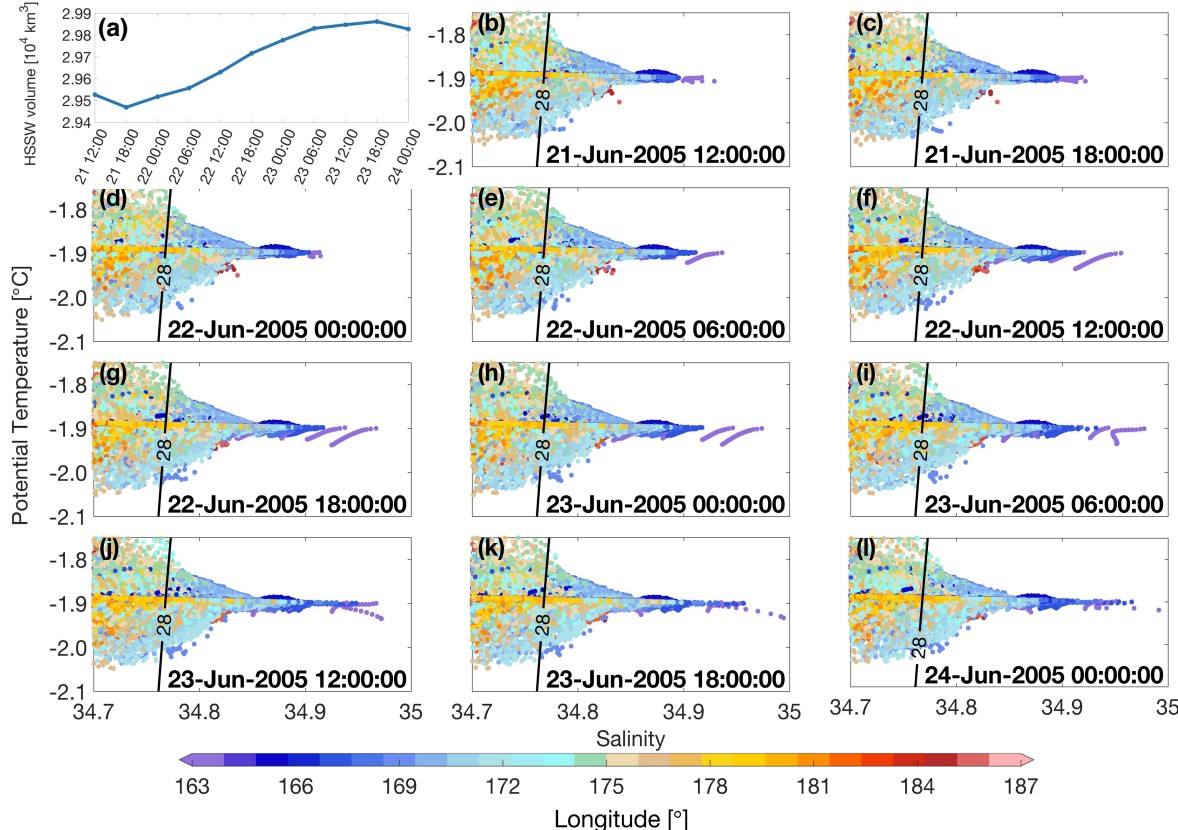

**Figure 10.** (a) Time series of HSSW volume over the RISP from 12:00 of June 21 to 00:00 of June 24 2005. (b–l) Temperature–salinity (T–S) diagrams for the RISP region shown in Fig. 2a. The T–S dots are color-coded with longitude. The black isoline denotes the potential density contour of 28 kg m⁻³. T–S diagrams are bounded by the purple box shown in Fig. 6a and represent values during the MESO event from June 21–24 of 2005.

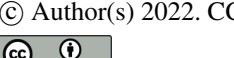



**Figure 11.** (a) Time series of averaged meridional winds along the S1 transect (see Fig. 1), HSSW exports across the S1 and averaged current velocity along the S1 from 18:00 of June 20 to 18:00 of June 25 2005. The gray shading represents the time of the MESO event. The correlation coefficient R and P-value were calculated between the HSSW export and meridional winds. (b) Same as Fig. 11a but for S3. (c–n) Vertical sections of cross-transect current velocity (color shading) and potential density (contour lines) on (c, f, i and l) S1, (d, g, j and m) S2 and (e, h, k and n) S3 at four selected time moments (indicated by the magenta triangles in (a) and (b)). Positive values denote currents from the RISP toward the open ocean, i.e. westward for S1 and northward for S2 and S3. The bold black contour line indicates the isopycnal of 28 kg m$^{-3}$.





The HSSW exports across the transects S1–S3 over the period of the mesoscale cyclone MESO are shown
in Fig. 11. To better investigate the relationship between the HSSW export and the cyclone event,
distributions of wind vectors and sea level pressure were examined for about two days after the end of
MESO (i.e. until June 25). The two maxima of wind speed time series are associated with two consecutive
mesoscale cyclones (Figs. 11a–b). For Transect S1, there is a significant, non-lagged negative correlation
between the meridional wind speed and HSSW export from June 21 to June 25 (R=-0.59, P=0.005) (Fig.
11a). However, relationships of wind speed and HSSW export across S1 and S3 are different. The HSSW
export across S3 is significantly and positively related to the wind speed with a 12-hour lag (R=0.90,
P<0.0001). The export across S2 has a negative correlation with wind speed, though weaker compared to
S1 (R=-0.44 and P=0.05, not shown). For transects S1 and S3, the current velocity is significantly and
positively correlated with the HSSW export with no lag (R>0.96 and P<0.0001,  Figs. 11a–b), resembling
the SYNO1 case. As there is a 12-hour lag correlation between wind speed and current velocity for S1,
such a relationship can explain why the HSSW export also exhibited a 12-hour-lag response to the wind
speed for S1. The HSSW export decreased by 57% across S1 and increased by 1.69 Sv across S3 when the
offshore wind speed increased from -3.3 to 7.1 m s$^{-1}$ compared to the values before MESO (Table 1).
Vertical sections for ocean currents and potential density are presented in Figs. 11c–n. For S1, positive
(westward) transports were observed in the upper 50 m (Figs. 11f and l) when offshore wind speed became
stronger, which can be interpreted as enhanced westward Ekman transport (Fig. 11a). Consistent with the
features found under synoptic-scale cyclones, the major factor inducing the change in HSSW export across
S1 is the current velocity change below 50 m: the westward velocity increased between 77.1–76.9°S and
76.7–76.5°S corresponding to the increase of wind speed, and the eastward velocity decreased in the central
section between 76.9–76.7°S (Figs. 11c to f and i to l), ultimately leading to a negative correlation between
the wind speed and the HSSW export. For S2, the most eminent feature is that the export of HSSW is
mainly concentrated on the eastern side of the section (Figs. 11d, g, j and m). For S3, there was no
significant change in HSSW compared to the change in current velocity. The sharp change in current
velocity around 178.4°E dominates the variation of HSSW export (Figs. 11e, h, k and n).
**3.4 Potential mechanisms for the HSSW export variations**
As illustrated in Section 3, for both SYNO1 and MESO, the HSSW export across S1 was negatively
correlated with the meridional wind speed, while export across S3 was positively correlated (Figs. 7a–b
and Figs. 11a–b). These relationships also exist over the longer time periods of June–September in 2005
and 2014 but with relatively lower correlation coefficients (R=-0.27, P<0.0001 with a lag of 6–12 hours for
S1 and R=0.42, P<0.0001 with a 12-hour lag for S3). Meanwhile, it is clear that the current velocity is the
dominant factor in modulating the HSSW export change compared to the HSSW volume for both S1 and
S3 (Figs. 7 and 11). Then, to elucidate the dynamical control for the current variations, we examined the
momentum budgets for S1 and S3 in SYNO1 and MESO. For SYNO1, the momentum balance presents
similar results to those of MESO and is not shown. The vertical sections of momentum terms on S1 and S3
in MESO are displayed in the cross-shelf (Fig. 12) and along-shelf (Fig. 13) directions respectively. For
both directions, the momentum budgets were dominated by the Coriolis acceleration and pressure gradient
terms, and the other terms were an order of magnitude smaller (Figs. 12 and 13). As such, the flows across
these transects (in the along-shelf direction for S1 and the cross-shelf direction for S3) were primarily
geostrophic (Figs. 12 and 13).
For S1, there were two zones (77.1–76.9°S and 76.7–76.5°S) where the current velocity changed notably
(Figs. 11c, f, i and l) during MESO, corresponding to the change in the Coriolis (Figs. 12b, g, l and q) term,
which was associated with the change in the pressure gradient (Figs. 12c, h, m and r) term. We then
examined the spatial distribution of sea surface elevation over the Ross Sea (Fig. 14), where negative values





near the RIS close to S1 existed persistently. Such distribution resulted in southward (negative) pressure
gradient force due to sea surface differences over S1, leading to an eastward geostrophic flow across S1.
When the wind speed increased at 12:00 of June 22 and 06:00 of June 24 (Fig. 11a), the pressure gradient
force was larger than that under lower wind speeds (compare Figs. 14b and 14d to 14a and 14c). These
features suggest that the increased offshore winds induced intensified westward Ekman transports in the
upper layer over 74–76.5°S and 163–173°E just north of S1 (Fig. S4), eventually resulting in the higher sea
surface in this region (Figs. 14b and 14d). Meanwhile, the relatively strong vertical shear in the upper layer
suggests that the Ekman transport could dominate on top of the interior geostrophic current (Figs. 12j and
12t). However, near the RIS over 163–173°E where offshore winds also prevailed, the increase in surface
elevation could barely be detected. After examining the horizontal pattern of currents over the Ross Sea,
we found a current flowing from west to east across S1 located north of the Ross Island, which further
flowed southward below the RIS (Fig. S4) in the upper layer. Previous studies showed that there is an
HSSW inflow underneath the RIS through the Ross Island (Budillon et al. 2003), which could advect more
than 10% of HSSW to the southern part of RIS, and intensify continuously over the winter (Jendersie et al.
2018). Therefore, we speculate this southward inflow as one of the reasons for the persistent low sea surface
elevation in the area close to the RIS (Fig. 14).

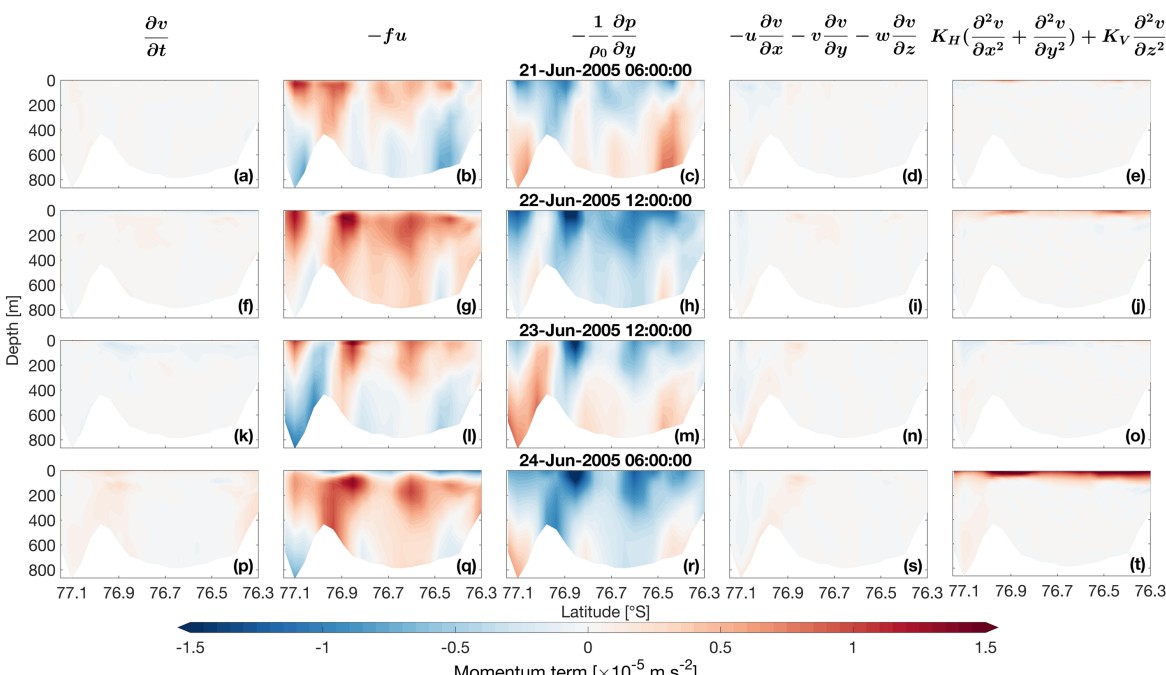

**Figure 12.** (a–t) Vertical sections of the momentum equation terms (Eq. (2)) along S1 at four selected time
moments during the MESO event (indicated by the magenta triangles in Figs. 11a and 11b): (a, f, k and p)
local acceleration, (b, g, l and q) Coriolis acceleration, (c, h, m and r) pressure gradient, (d, i, n and s)
nonlinear advection and (e, j, o and t) eddy viscosity in the cross-shelf momentum budget.

Along S3, the western section (around 178.4°E) with a considerable velocity change (Figs. 11e, h, k and n)
dominated the variation of HSSW export. Meanwhile, an outward (northward) flow can be seen clearly
over the Glomar Challenger Trough across the western section (marked by the red box in Fig. S4). From
the distribution of sea surface elevation, it is noted that the elevation was lower when the wind speed





increased over the Glomar Challenger Trough (Figs. 14b and 14d), which might be associated with the
divergent Ekman transports caused by the cyclone. Such a divergent pattern would generate a positive
(eastward) pressure gradient force over the western section of S3, which drove northward geostrophic flows
associated with the HSSW transport occupying this area (Figs. 11e, h, k and n).

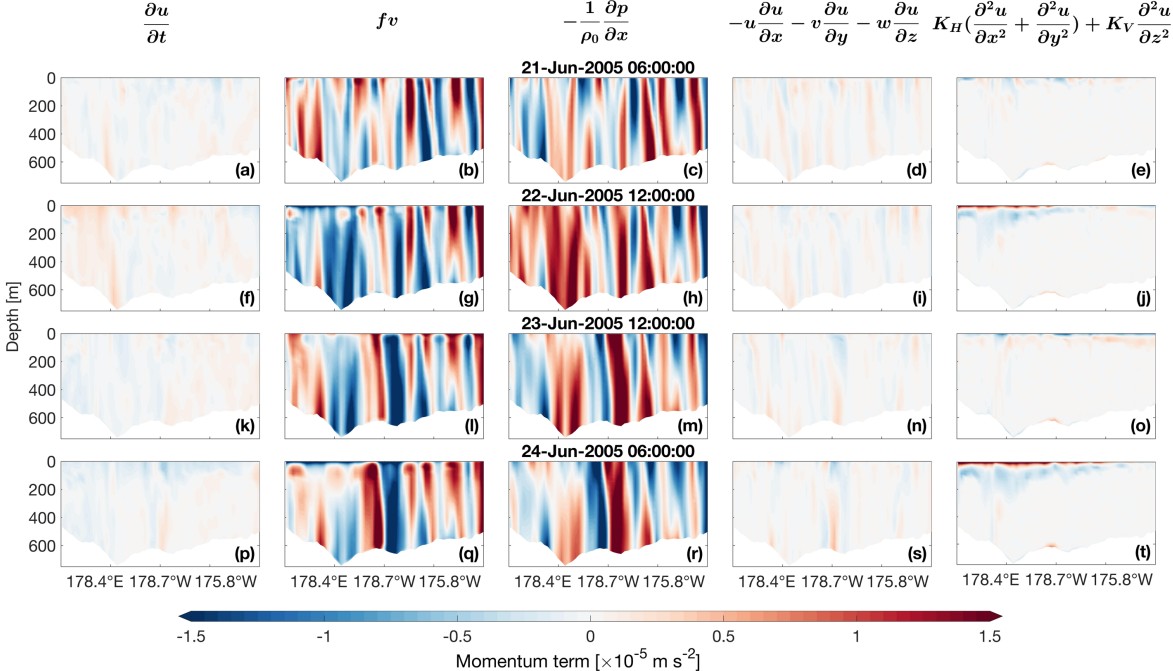


**Figure 13.** (a–t) Vertical sections of the momentum equation terms (Eq. (1)) along S3 at four selected time
moments during the MESO event (indicated by the magenta triangles in Figs. 11a and 11b): (a, f, k and p)
local acceleration, (b, g, l and q) Coriolis acceleration, (c, h, m and r) pressure gradient force, (d, i, n and s)
nonlinear advection and (e, j, o and t) eddy viscosity in the along-shelf momentum budget.



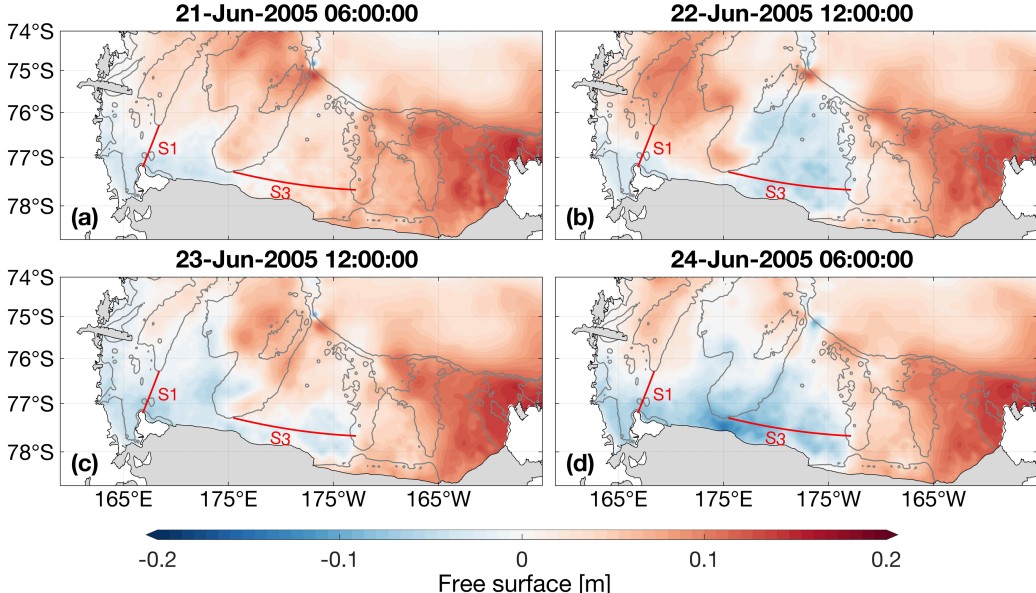


**Figure 14.** (a–d) Spatial distributions of free surface in the Ross Sea region at four selected time moments
of MESO (indicated by the magenta triangles in Figs. 11a and 11b). The red lines indicate the S1 and S3
section, and the gray lines indicate 500 and 1000 m isobaths.


3.5 Lag time for HSSW formation and export
The HSSW formation in the RISP demonstrated a near-instantaneous response to the wind change during
the synoptic- and meso-scale cyclone events (Figs. 6, S3 and 10), which could persist for 12–48 hours after
the passage of the cyclones. These features are somewhat different from the HSSW response over the East
Antarctica coastal polynyas as proposed by Wang et al. (2021), which elucidates a lag response of 10–15
days for the HSSW formation to strong wind events in the Prydz Bay and Shackleton polynyas. Such
discrepancy might be related to the polynya extent and local circulations. The RISP has been regarded as
the highest ice production region among the major 13 Antarctic coastal polynyas (Tamura et al., 2008),
suggesting intensified brine rejection that will result in faster production of HSSW. Another factor might
be the local circulation system including the outflow of basal melting water and the circumpolar deep water
(CDW) intrusion processes over these coastal regions. The CDW intrusions and HSSW exports are highly
spatially and temporally correlated on subdaily time scales over the Ross Sea (Morrison et al., 2020), so
the cross-slope transports of CDW could largely modulate the HSSW transport in canyons. Meanwhile, the
increased freshwater from ice shelf melting could pump more CDW into the ice shelf cavities (Jourdain et
al. 2017) and reduce the transport of CDW onto the continental shelf region (Dinniman et al. 2018), which
can further affect the formation of dense shelf water.

The HSSW export across S3 was positively correlated with wind speed over MESO with a 12-hour lag (Fig.
11b). Furthermore, as mentioned before, such a lag relationship between wind and HSSW was robust over
June–September in 2005 and 2014, while the lag time could vary between 6 hours and 12 hours. Mathiot
et al. (2012) documented a 6-month time lag between the HSSW formation in polynyas (TNB and RISP)
and the HSSW transport across the topographic sills in the Ross Sea. The defined sections across the





Drygalski Trough and the Joides Trough in their study were located around 74°S, which is about 330 km
further north than the sections we selected. The lag time between the changes in wind and HSSW exports
is highly dependent on the locations of chosen transects and the spreading rate of HSSW.

**4 Conclusions**
This study investigated the response of sea ice and HSSW formation and export in the Ross Ice Shelf
Polynya to meso- and synoptic-scale cyclones based on a coupled ocean-sea ice-ice shelf Ross Sea model.
For synoptic- and meso-scale cyclones, two and one representative events were respectively selected. When
synoptic-scale cyclones prevailed over this region, the entire RISP was dominated by strong offshore winds,
which resulted in increased SIP rates. During the passage of the mesoscale cyclone, SIP increased rapidly
over the western side of RISP but decreased over the eastern side of RISP, due to changes in the offshore
winds associated with the cyclonic wind field. SIP instantaneously responded to the wind change over the
RISP under both the synoptic-scale and mesoscale cyclones. Enhanced HSSW formation was detected
when there was a notable increase of SIP in RISP, mainly in the western side of RISP, and could persist for
12–48 hours after the passage of the cyclones. For the two synoptic-scale cyclones, the increase in HSSW
formation persisted for about 2 days, while the response of HSSW formation to the mesoscale cyclone had
a shorter lag of about 12–18 hours. The HSSW export across the transect over the Drygalski Trough (S1)
negatively correlated with the meridional wind, while the export across the transect over the Glomar
Challenger Trough (S3) was positively correlated with the wind. The variations of the HSSW export across
S1 and S3 were mainly regulated by the geostrophic currents. Pressure gradients driving the geostrophic
currents were related to gradients in sea surface caused by wind-induced Ekman transports. However, there
might be other factors that affected the hydrography near the Ross Island along the S1 transect. For instance,
the melting beneath the RIS and the intrusion of the circumpolar deep water have impacts on currents in
this region, which deserves future investigations to reveal the different responses for S1 and S3. In addition,
tides could further modulate the export and volume of HSSW in this region (Padman et al. 2009; Wang et
al. 2013), and such effects should be considered by using models including the tides in the future.


Data availability.
The model data that support the findings of this study are available at
https://www.dropbox.com/sh/9ilxwsft080cdds/AABmnhFaMKRu2XL98C4YysXDa?dl=0. More details
about other observed data are presented in Sect. 2.

Author contributions.
ZZ and XW designed the original ideas presented in this manuscript. ZZ conceived the project of response
of Antarctic coastal polynya processes to strong wind events funded by the Shanghai Science and
Technology Committee. XW conducted the model simulation analysis. W and ZZ wrote the original
manuscript draft. MD conducted the 5-day-average model simulations and XW conducted the 6-hourly
outputs. MD, PU, XL and MZ participated in the result interpretation, manuscript preparation and
improvement. All authors contributed to the article and approved the submitted version.

Competing interests. The authors declare that there is no conflict of interest.



Acknowledgements.
This work is funded by the National Natural Science Foundation of China (Grant No. 41941008 and
41876221), the Shanghai Science and Technology Committee (Grant No. 20230711100 and Grant No.
21QA1404300), the Impact and Response of Antarctic Seas to Climate Change (Grant 583 No: IRASCC
1-02-01B), the National Key Research and Development Program of China (Grant No. 2019YFC1509102),
the Shanghai Pilot Program for Basic Research - Shanghai Jiao Tong University, and the Shanghai Frontiers
Science Center of Polar (SCOPS). Work of PU and ZZ was supported by the European Union's Horizon
2020 research and innovation framework programme under Grant agreement no. 101003590 (PolarRES
project). PU was also supported by the Academy of Finland (Project 322432).



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
