# Peer review of "The response of sea ice and high salinity shelf water in the Ross Ice Shelf Polynya to cyclonic"

_The Cryosphere, 2022_

## Author Comment (AC1)

Response to Review Comments

for

**"The response of sea ice and high salinity shelf water in the Ross Ice Shelf Polynya to cyclonic atmosphere circulations"**

Xiaoqiao Wang, Zhaoru Zhang, Michael S. Dinniman, Petteri Uotila, Xichen Li, Meng Zhou

**Note**: Reviewers' comments are in italic font; authors' response comments are in normal font. Revisions in the revised manuscript are highlighted by blue color.

**Reviewer comments:**

*Anonymous Referee #1:*

*Overview Comments:*

*This paper is a novel work discussing the formation and exportation of HSSW with the influence of cyclones. The cyclones leading to the extreme winds event can enhance the formation and outflowing of HSSW, while as the authors point out that the previous studies always focus on the influence of seasonal scale winds such as katabatic winds. This paper enriches our understanding in this area. It is very interesting that cyclones contribute to outward barotropic currents by changing sea surface height and thus affect HSSW exportation.*

*However, some discussions fall short in view of the possible implications, which are listed below.*

We thank the reviewer for the considerable efforts in reviewing this manuscript. The useful and helpful comments helped to clarify and significantly improve the manuscript. Please see our response to each comment as follows.

*1. In this paper, three cases (SYNO1, SYNO2 and MESO) are shown. However, only SYNO1 and MESO are analyzed in detailed. The authors may consider SYNO2 to be similar to SYNO1, and the analysis of SYNO1 could be also applied to SYNO2. However, there are some differences between SYNO2 and SYNO1, and the reasons for such differences have not been fully discussed. For example, there is a weak correlation between the meridional winds and the HSSW exportation across S1 in SYNO2, while in SYNO1 and even MESO, the negative correlation is significant. Thus, we are not sure that the conclusions got from SYNO1 and MESO can be also applied to SYNO2. If the authors want to use the case of SYNO2 to show that the conclusion in this paper can be generalized, it is necessary to discuss the differences between SYNO1 and SYNO2, otherwise people may doubt the applicability of the conclusion.*

Sorry for the lack of discussion on this part. For the SYNO2 event, the main reason for the weak correlation between HSSW export and winds should be related to the cyclone center, which is located on the eastern side of the Ross Sea close to the Amundsen Sea (Fig. S1 in the supplementary), leading to lower wind speeds in the RISP region than the other two events (SYNO1 and MESO). As can be seen from Table 1 in the revised manuscript, in the RISP region the mean wind speed for the SYNO2 event is only about 5.3 m s$^{-1}$, but for SYNO1 and MESO, the mean wind speed is greater than 7 m s$^{-1}$. In addition to the lower wind speed, there may be other factors (such as under ice shelf circulations) that play a

significant role in regulating HSSW export, which makes the role of wind speed weakened. The related discussion has been added "The reason for the weaker correlations between HSSW export and wind speed weover SYNO2 might be related to the lower wind speed in RISP than over SYNO1 event (5.3 m s$^{-1}$ for SYNO2, 7.0 m s$^{-1}$ for SYNO1, shown in Table 1), resulting from the faraway cyclone center located in the Amundsen Sea. Additionally, other factors (such as ice shelf circulations) could regulate the HSSW exports significantly." (Lines 393–397 in the revised manuscript).

*2. The authors set three sections (S1, S2, S3) to describe the exportation of HSSW. However, this paper mainly focuses on the HSSW exporting across S1 and S3 and ignores the S2. I don't think the phenomenon on S2 is trivial, although the correlation between the meridional winds and the HSSW across S2 is not as significant as the cases of S1 and S3, which is not consistent with expectations. But it will also be interesting, if the authors can explain why the correlation is weak.*

By examining the spatial pattern of horizontal currents near S2 (Figs. 15 and S7 in the revised manuscript), there is a strong northward flow across S2 at deeper layers (300 m and 500 m) near 175°E (revised Figs. 15e, h, k and f, i, l). This flow originated around 79°S which is located at the RIS (revised Figs. S7e, h, k and f, i, l), so it could be an interaction between basal melting water and the HSSW. Some previous studies have already demonstrated significant effects of ice shelf water on HSSW formation and export (Herraiz-Borreguero et al., 2016; Williams et al., 2016). Accordingly, the weaker correlation between HSSW export and wind speed for S2 might be associated with these ice shelf circulations. This possible explanation has been added as "By examining the ocean currents near S2, a northward flow originated around 79°S which is located at the RIS (revised Figs. S7e, h, k and f, i, l) was observed, and the weaker correlation between HSSW export and wind speed might be associated with local ice shelf circulations." (Lines 477–479 in the revised manuscript).

References:
Herraiz-Borreguero, L., J. A. Church, I. Allison, B. Pen~a-Molino, R. Coleman, M. Tomczak, and M. Craven: Basal melt, seasonal water mass transformation, ocean current variability, and deep convection processes along the Amery Ice Shelf calving front, East Antarctica, J. Geophys. Res. Oceans, 121, 4946–4965, doi:10.1002/2016JC011858, 2016.
Williams, G. D., Herraiz-Borreguero, L., Roquet, F., Tamura, T., Ohshima, K. I., Fukamachi, Y., Fraser, A. D., Gao, L., Chen, H., McMahon, C. R., Harcourt, R., and Hindell, M.: The suppression of Antarctic bottom water formation by melting ice shelves in Prydz Bay, Nat. Commun., 7, https://doi.org/10.1038/ncomms12577, 2016.

*Specific Comments:*

*Following, there is a number of specific comments, many of which can be considered as minor.*

*3. Line 39-41 ('In general, ... synoptic-scale atmosphere forcing'): This sentence is similar with that of Line 36. May be them can be merged.*

Following the reviewer's suggestion, the original sentence "In general, the near-surface wind fields over the Antarctic continent are forced by both katabatic flow and meso- or synoptic-scale atmospheric forcing." has been merge with "These coastal polynyas are mechanically driven by offshore katabatic and synoptic winds" in Line 36, and finally revised to "These coastal polynyas are mechanically driven by offshore katabatic and synoptic winds (Bromwich et al., 1998; Massom et al., 1998; Morales Maqueda et al., 2004), which are regarded as the dominant near-surface wind fields over the Antarctic continent." (Lines 36–38 in the revised manuscript).

*4. Line 63: the 'recent studies' need some Refs.*

The related references have been added here and the original sentence has been revised to "Recent studies have… based on the observations (Dale et al., 2017; Cheng et al., 2019; Ding et al., 2020; Thompson et al., 2020; Wenta and Cassano., 2020)." (Lines 62–65 in the revised version).

*5. Line 52: 'sea ice production (SIP) rate' is correct.*

"sea ice production rate (SIP)" has been replaced by "sea ice production (SIP) rate".

*6. Line 66-70: is the Thompson et al., (2020) shown to indicate there is a strong response of coastal polynya SIP to the extreme winds event? If so, I think you should tell the readers the magnitude of the mean SIP.*

Sorry for the unclear writing. The value of 110 cm d$^{-1}$ in the original sentence (Line 70) is the magnitude of mean SIP found by Thompson et al. (2020). Then this sentence "Thompson et al. (2020) demonstrated using in-situ observations, available from the Polynyas and Ice Production and seasonal Evolution in the Ross Sea (PIPERS) program which conducted an autumn ship campaign in 2017 and two spring airborne campaigns in 2016 and 2017, that the estimated frazil ice production could increase up to 110 cm d$^{-1}$ during the strongest wind events (Ackley et al., 2020)." has been revised to "Thompson et al. (2020) demonstrated that the estimated frazil ice production could increase up to 110 cm d$^{-1}$ during the strongest wind events using in-situ observations, available from the Polynyas and Ice Production and seasonal Evolution in the Ross Sea (PIPERS) program, which conducted an autumn ship campaign in 2017 and two spring airborne campaigns in 2016 and 2017 (Ackley et al., 2020)" to avoid such confusion (Lines 66–70 in the revised version).

*7. Line 71: 'Terra Nova Bay (TNB) polynya' may be a clearer statement.*

"Terra Nova Bay polynya (TNB)" has been changed to "Terra Nova Bay (TNB) polynya".

*8. Line 121: why does only climatological precipitation data come from AMPS, while the rest meteorological factors come from ERA-Interim? The AMPS data covers 2008-2022, which ends much later than the study period (1999-2014). Moreover, the temporal resolution of precipitation, sea level pressure and humidity are different from that of winds and air temperature. Maybe its influence is not significant, or there are some other reasons, but it needs to be claimed.*

The main reason for using climatological precipitation from AMPS is related to the performance of coastal precipitation around Antarctica, which is significantly affected by the atmospheric model resolution (van Lipzig et al., 2004). The AMPS product has high-resolution atmospheric forecast fields for Antarctica computed from a mesoscale meteorological model. Therefore, a monthly climatology of precipitation from AMPS was used in our model instead of the ERA-Interim precipitation. Meanwhile, this regional Ross Sea model covered from 1999–2014 was developed in 2015 and we planned on 15-year simulations at that time, so the ERA-Interim product becomes the best choice at that moment. The high temporal resolution for winds and air temperature used in our model is related to their importance in simulating sea ice and currents over the Southern Ocean. For all the other atmospheric forcing, the temporal resolution does not have significant impacts on the model outputs. In addition, Wu et al. (2020) further proposed that the influence of atmospheric variability in the Southern Ocean mainly results from wind fluctuations.

The related details have been added as "Coastal precipitation from reanalysis products for Antarctica is significantly affected by atmospheric model resolution (van Lipzig et al., 2004). Therefore, monthly climatological precipitation used in this model is derived from the Antarctic Mesoscale Prediction System (AMPS), a high-resolution atmospheric model over the Antarctic (Powers et al., 2003; Bromwich et al.,

2005), instead of the ERA-Interim product. Furthermore, due to the overestimation of mean clouds over the Southern Ocean from ERA-Interim, monthly cloud fraction climatology data comes from the International Satellite Cloud Climatology Project stage D2…" (Lines 125–131 in the revised version) and "The high temporal resolution for winds and air temperature is related to their importance for simulating sea ice and currents in the Southern Ocean (Wu et al., 2020)." (Lines 123–125 in the revised version).

References:
van Lipzig, N. P. M., King, J. C., Lachlan-Cope, T. A., and van den Broeke, M. R.: Precipitation, sublimation, and snow drift in the Antarctic Peninsula region from a regional atmospheric model, 109, 1–16, https://doi.org/10.1029/2004JD004701, 2004.
Wu, Y., Wang, Z., Liu, C., and Lin, X.: Impacts of High-Frequency Atmospheric Forcing on Southern Ocean Circulation and Antarctic Sea Ice, 37, 515–531, https://doi.org/10.1007/s00376-020-9203-x, 2020.

*9. Line 167: how to calculate the SIP in RISP? The mean SIP in the RISP or the in the RISP Polynya region?*

Following Nihashi and Ohshima (2015), the calculation of annual cumulative SIP covers the period of March to October, i.e. the ice freezing seasons around Antarctica. Therefore, we calculated the annual cumulative SIP averaged over 2003–2010 (The temporal coverage of satellite-retrieved data) based on this method instead of polynya-mean values for RISP. To illustrate the calculation more clearly, we have added this sentence "Following Nihashi and Ohshima (2015), the calculation of annual cumulative SIP covers the period of March to October, i.e. the ice freezing seasons around Antarctica." (Lines 172–173 in the revised version).

References:
Nihashi, S. and Ohshima, K. I.: Circumpolar Mapping of Antarctic Coastal Polynyas and Landfast Sea Ice: Relationship and Variability, J. Clim., 28, 3650–3670, https://doi.org/10.1175/JCLI-D-14-00369.1, 2015.

*10. Line 189: the extent of RISP may be overestimated due to the zero SIP of threshold. The regions covered by thicker sea ice may also produce ice.*

To clarify the definition for the extent of RISP, we modified this sentence "The extent of RISP was defined as the area where the multi-year-average annual cumulative SIP is greater than zero based on the Ross Sea model results." as "The extent of RISP was defined as the area where the multi-year-average annual cumulative SIP is greater than zero near the RIS region based on the Ross Sea model results." (Lines 195–196 in the revised version). By examining the spatial distribution of sea ice thickness (SIT) and sea ice concentration (SIC) as shown by Figs. R1 and R2 respectively, the coastal polynya region is characterized by low SIT and SIC values over the SYNO1. These similar features could also be found during SYNO2 and MESO, therefore the extent of RISP we defined before could not be affected by thicker sea ice over the coastal region.

[Figure]

**Figure R1**. Spatial distributions of 12-hour-average sea ice thickness in the Ross Ice Shelf Polynya over 13–17 July 2005.

[Figure]

**Figure R2**. Spatial distributions of 12-hour-average sea ice concentration in the Ross Ice Shelf Polynya over 13–17 July 2005.

*11. Line 192: the RISP polygon region comprises the McMurdo polynya. Although this polynya is small and without strong SIP, I think it will be better to limit the western boundary to Ross Island to exclude the McMurdo polynya.*

Sorry for the confusion associated with the defined RISP polygon region. As the reviewer suggested, we redefined the boundary of RISP to make a better separation between McMurdo polynya and RISP. The western boundary has been revised close to Ross Island shown in Fig. 2a in the revised manuscript. Furthermore, all related results have been updated based on this newly defined extent.

*12. Line 240: maybe claiming the name of case (e.g., SYNO1) here can make the reader clearer.*

Following the reviewer's suggestion, the original sentence "For synoptic-scale cyclones, two events were selected for our study, which occurred in July 2005 and September 2014, respectively." has been revised to "For synoptic-scale cyclones, two events were selected for our study, which occurred in July 2005 (SYNO1) and September 2014 (SYNO2)" (Lines 249–250 in the revised version).

*13. Line 242: plot the cyclone tracks of the cases on Figure 3. I think you want to show that the cyclones chose here is typical. You can claim it at the end of this paragraph.*

The cyclone tracks of three typical cases (SYNO1, SYNO2 and MESO) are added to Fig. 3 as the reviewer suggested, and descriptions of the tracks and cyclone evolutions are added in the revised version (Lines 250–261).

*14. Line 258: Can extend the case of SYNO1 forward to when the cyclone did not reach the Ross Sea? It will make the impact of cyclones even more significant.*

When selecting SYNO1, we presented the spatial distributions for all related variables from June to September to define the most proper period of SYNO1. Two days before the onset of SYNO1, several low-pressure systems were visible by examining the spatial pattern of sea level pressure as shown in Fig. R3. These cyclones on the north of the Ross Sea gradually merged, which eventually became SYNO1 we have selected. Therefore, if the defined time range is extended forward, other cyclonic processes associated with early development for SYNO1 might be included. But in this study, we only focus on the more mature stage of the cyclone, in other words, when the cyclone has a larger spatial scale, stronger intensity and only one low-pressure center. In addition, the changes in sea ice production in RISP were not significant especially for the western part (Fig. R4), indicating these earlier atmospheric processes could not affect the polynya activities notably. So based on these characteristics, we finally defined 13–17 July as the time range for SYNO1. The main reason has been added as "The SYNO1 occurred from July 13 to July 17 of 2005, when the cyclone was situated in a mature stage, i.e. when the cyclone has a large spatial scale, strong intensity and only one low-pressure center." in the revised version (Lines 272–274).

[Figure]

**Figure R3**. Spatial distributions of 12-hour-average sea level pressure (color shading) and 10-m wind vectors (black arrows) in the Ross Ice Shelf Polynya over 11–13 July 2005.

[Figure]

**Figure R4**. Spatial distributions of 12-hour-average sea ice production in the Ross Ice Shelf Polynya over 11–13 July 2005.

*15. Line 301: the Figure 6n shows the HSSW volume, which can reflect the accumulation of density water. However, the cyclones directly affect HSSW generation (i.e., increasing). Thus, the increasing of HSSW may be a more suitable factor, which should be shown here. Furthermore, you said that 'the HSSW volume ... reached the maximum at 06:00 of July 17', but after July 17, HSSW volume still increases. I think here you want to say that the rapid increase of HSSW volume ends at 06:00 of July 17, which can also be illustrated significantly with the plot of increasing rate of HSSW volume.*

It is true that this sentence aims to proposed that the increase of HSSW volume ends at 06:00 of July 17. Therefore, it is clearer to show the variation of the increase in HSSW volume as the reviewer suggested. The HSSW volume has been changed to the variability of HSSW volume shown in Figs. 6, 10 and S3. The related content has been revised according to these updated figures as "The HSSW volume increased apparently from 18:00 of July 14 to 06:00 of July 16 (Fig. 6b), when a dramatic increase in SIP occurred in this area accompanied by the intensification of the SYNO1 cyclone. The HSSW volume still kept increasing indicted by the positive values for HSSW volume variability even when SYNO1 weakened from 18:00 of July 16 to 18:00 of July 19 (Fig. 6b), so the HSSW formation could persist around 3 days after the cyclone decayed. For the SYNO1 event, the HSSW volume increased by 0.06·104 km3 compared to the value before the cyclone (Table 1). Meanwhile, the HSSW salinity presented similar features with HSSW volume variability and reached to the maximum at 18:00 of July 16 (Fig. 6c) when

the cyclone had intensified for 2 days. The higher-salinity HSSW persisted for 2-3 days after the decay of the cyclone from 18:00 of July 16 to 06:00 of July 19, and then the salinity started decreasing (Figs. 6c)." (Lines 309–318 in the revised version).

In addition, the HSSW is redefined as the water mass with neutral density ($\gamma^n$) above 28.27 kg m$^{-3}$, salinity ($S$) $> 34.62$ and potential temperature ($\theta$) $< -1.85°$ C (Orsi and Wiederwohl, 2009; Castagno et al., 2019) in the revised version based on the second reviewer's suggestion.

References:
Castagno, P., Capozzi, V., DiTullio, G. R., Falco, P., Fusco, G., Rintoul, S. R., Spezie, G., and Budillon, G.: Rebound of shelf water salinity in the Ross Sea, 10, 1–6, https://doi.org/10.1038/s41467-019-13083-8, 2019.
Orsi, A. H. and Wiederwohl, C. L.: A recount of Ross Sea waters, Deep. Res. Part II Top. Stud. Oceanogr., 56, 778–795, https://doi.org/10.1016/j.dsr2.2008.10.033, 2009.

*16. Line 309: may be 'eastward (negative, toward RISP)' can be clearer.*

Sorry for this confusing definition about directions. The definition of along-shelf velocity sign in the original manuscript is indeed not common in the oceanography community, though we wanted to define the direction of HSSW export toward the slope (westward) as positive. As the reviewer suggested, we changed the definition and now the eastward direction is defined positive in the revised manuscript. All related texts and figures (revised Figs. 7 and 11) have been updated using this new definition.

This original sentence "The export of HSSW across S1 has a significant negative correlation with the meridional wind speed (-R=0.70, P=0.012), suggesting that the HSSW had stronger eastward (negative) transport across the meridionally directed transect S1 when the wind speed increased." has been revised to "The export of HSSW across S1 has a significant positive correlation with the meridional wind speed (R=0.70, P=0.012), suggesting that the HSSW had stronger eastward (positive, toward RISP) transport across the meridionally directed transect S1 when the wind speed increased." (Lines 331–333 in the revised version).

*17. Line 313: the sentence of '(R>0.98 and P<0.0001, not shown)' may indicate that the correlation coefficient is not shown in the figure. But it could be misleading that these two factors are not plotted. So, I think the 'not shown' can be deleted. Furthermore, this result would be interesting, if we can estimate that how much of the increase in HSSW exportation is due to the increase in current speed, and how much is due to the increase in HSSW volume (area in the section)?*

Following the reviewer's suggestion, the original sentence "The averaged current velocity on S1 and S3 both have strong positive correlations with HSSW export (R>0.98 and P<0.0001, not shown), suggesting…" has been revised to "The averaged current velocity on S1 and S3 both have strong correlations with HSSW export ($R^2$>0.98 and P<0.0001), suggesting…" (Lines 335–336 in the revised version).

Time series of HSSW exports across the defined transects (S1 and S3) and averaged current velocity along these transects has been presented in Figs.R5 and R6 for SYNO1 and MESO respectively. Similarly, Figs.R7 and R8 show the time series of HSSW exports and HSSW amount (i.e., HSSW volume) for S1 and S3 over SYNO1 and MESO events. Obviously, the HSSW export significantly correlated with the current velocity for both S1 and S3 ($R^2$>0.98 and P<0.0001, Figs. R5 and R6). However, there are no consistent correlations between HSSW export and HSSW volume for S1 and S3. The correlation coefficients between HSSW export and volume are above 0.7 (P<0.01) for S1 in SYNO1 and S3 in MESO (Figs. R7a and R8b), but for the other cases the correlations are weaker (Fig. R8a) or even turn to the

negative value (Fig. R7b) which might be associated with the persistence of HSSW formation. So generally, the current velocity is the main factor in regulating the HSSW export.

[Figure]

**Figure R5**. (a) Time series of HSSW exports across the S1 and averaged current velocity along the S1 from 06:00 of July 13 to 18:00 of July 18 2005. The gray shading represents the time of the SYNO1 event. The correlation coefficient R and P-value were calculated between the HSSW export and current velocity. (b) Same as Fig. R5a but for S3.

[Figure]

**Figure R6**. (a) Time series of HSSW exports across the S1 and averaged current velocity along the S1 from 18:00 of June 20 to 18:00 of June 25 2005. The gray shading represents the time of the MESO event. The correlation coefficient R and P-value were calculated between the HSSW export and current velocity. (b) Same as Fig. R6a but for S3.

[Figure]

**Figure R7**. (a) Time series of HSSW exports across the S1 and HSSW amount along the S1 from 18:00 of June 20 to 18:00 of June 25 2005. The gray shading represents the time of the MESO event. The

correlation coefficient R and P-value were calculated between the HSSW export and HSSW amount. (b) Same as Fig. R7a but for S3.

[Figure]

**Figure R8**. (a) Time series of HSSW exports across the S1 and HSSW amount along the S1 from 18:00 of June 20 to 18:00 of June 25 2005. The gray shading represents the time of the MESO event. The correlation coefficient R and P-value were calculated between the HSSW export and HSSW amount. (b) Same as Fig. R8a but for S3.

*18. Line 371: Figure S3 shows that the high salinity water always appears at 163E, far away from the east side of RISP. So, it may be doubtful to attribute the slightly decreasing of HSSW salinity to SIP reduction at east side of RISP. Moreover, the Figure S3 g-j are not mentioned.*

The initial idea was that the SIP on the east side and the HSSW on the west side could be linked by the westward coastal current along the RIS (Pillsbury & Jacobs 1985; Jacobs & Giulivi 1998; Smith et al., 2012; Porter et al., 2019), which could be seen over the upper layers in Fig. 15 and Fig. S6 in the revised version (particularly significant in Figs. S6d, e, g, h, f and k). Such current might carry fresher water resulting from the decrease in SIP to the western RISP, which could eventually lead to the decrease of HSSW salinity. However, based on the suggestion from the second reviewer, we changed these T-S diagrams to the time series of HSSW salinity, so this part has been removed.

References:
Pillsbury, R. D., & Jacobs, S. S. (1985). Preliminary observations from long-term current meter moorings near the Ross Ice Shelf, Antarctica. In Oceanology of the Antarctic Continental Shelf, Antarctic Research Series (Vol. 43, pp. 87–107). Washington, D.C: American Geophysical Union. http://doi.org/10.1029/AR043p0087.
Jacobs, S. S., & Giulivi, C. F. (1998). Thermohaline Data and Ocean Circulation on the Ross Sea Continental Shelf. https://doi.org/10.1007/978-88-470-2250-8
Smith JR, W. O., Sedwick, P. N., Arrigo, K. R., Ainley, D. G., & Orsi, A. H. (2012). The Ross Sea in a sea of Change. Oceanography, 25(3), 90–103.
Porter, D. F., Springer, S. R., Padman, L., Fricker, H. A., Tinto, K. J., Riser, S. C., et al. (2019). Evolution of the seasonal surface mixed layer of the Ross Sea, Antarctica, observed with autonomous profiling floats. Journal of Geophysical Research: Oceans. https://doi.org/10.1029/2018JC014683.

*19. Line 376: the weak correlation between the HSSW exportation and meridional winds is interesting, which is different from SYNO1 and MESO. I think you should explain it.*

Please see our detailed response to Comment #1.

*20. Line 405-410: The estimation from Thompson et al. (2020) is obtained at TNB with an extreme katabatic wind. If you want to use it to verity the reliability of your model, maybe the wind speed, air temperature, etc. should be compared, which is important for SIP.*

Thanks for pointing out this issue. The maximum wind speed during the MESO event was around 15 m s$^{-1}$ (Figs. 11a and 11b in the revised version) and the 2-m air temperature was below −25°C during the period when the SIP reached its maximum value (indicated by the white boxes in Figs. R9). Therefore, the original sentence "Thompson et al. (2020) proposed that the intensity of ice production could rise up to 1.1 m day$^{-1}$ during the windiest events in TNB, which was calculated from the salt budget using conductivity–temperature–depth (CTD) profiles. Meanwhile, the maximum SIP rate in the RISP during MESO was 1.2 m day$^{-1}$ in our study, which is just slightly different from the value in TNB. Differences in topography and location between the RISP and TNB could lead to such slight differences in atmospheric and hydrological conditions." has been modified to "Thompson et al. (2020) proposed that the intensity of ice production could rise up to 1.1 m day$^{-1}$ during the events when the wind speeds exceeded 20 m s$^{-1}$ in TNB, which was calculated from the salt budget using conductivity–temperature–depth (CTD) profiles. Meanwhile, the maximum SIP rate in the RISP during MESO was 1.2 m day$^{-1}$ in our study when the wind speed was around 15 m s$^{-1}$, which is just slightly different from the value in TNB. The air temperatures for both TNB and RISP were below −25°C. Differences in topography and location between the RISP and TNB could lead to such slight differences in atmospheric and hydrological conditions." (Lines 428–434 in the revised version).

[Figure]

**Figure R9**. Spatial distributions of 6-hour-average 2-m air temperature in the Ross Ice Shelf Polynya over 21–23 June 2005. The white boxes represent the high sea ice production region during MESO shown in Figs. 9d and 9e.

*21. Line 429: The persistence is interesting. At 23-Jun-2005 06:00, the SIP of RISP is weak (Figure 9h), but the decreasing of HSSW salinity starts at 24-Jun-2005 24:00. Can you explain which process provide the salty water? The similar question also can be seen in SYNO1 (Line 297).*

The persistence of the higher-salinity HSSW when the SIP was decreasing might be associated with the brine rejection process. A certain time is required from the moment of new ice formation (i.e., SIP greater than zero) until the time when enough salinity is rejected to form the HSSW. Therefore, we speculate that the time required for the whole HSSW formation process may cause a lag in the response of HSSW salinity to SIP, which in turn leads to a continuous increase in salinity during the SIP reduction. Meanwhile, another possible reason might be related to the mixing process between different water masses. The cold and salty HSSW could be mixed with the fresher ambient water and the resulting water mass may also be in a highly-salinity state (i.e., still satisfy the definition of HSSW), which would then also lead to a continuous increase in HSSW salinity. The last possible reason is that the local circulation system might cause HSSW to be trapped in the polynya region for a period of time. To reveal the specific mechanism of this persistence for high-salinity HSSW, we will conduct sensitivity experiments and add online passive tracers to HSSW to identify the time scale required for the entire response process and the specific pathway of HSSW.

*22. Line 477: that's interesting. With the Coriolis force, the exportation of HSSW should concentrate on the left side of the trough (e.g., Wang et al., 2013), but here it is on the right side? Can you explain it?*

The export of HSSW concentrated on the eastern side of S2 (Figs. 11d, g, j and m in the revised version) originates from the Ross Ice Shelf region, which can be seen by looking at the revised Figs. 15 and S7. The presence of a northward flow across the eastern part of S2 at deeper layers (300 m and 500 m) can be clearly observed near 175°E (revised Figs. 15e, h, k and f, i, l). The flow originates around 79°S which is located at the RIS (revised Figs. S7e, h, k and f, i, l), so the flow could be an interaction between basal melting water and the HSSW. However, for the features revealed in Wang et al. (2013), their model domain as far south as 77°S does not encompass the Ross Ice Shelf, so the influence of ice shelf processes on the HSSW transport cannot be portrayed. Therefore, we speculate that the ice shelf processes are the main reason for such differences. The related information has been added as "For S2, the most eminent feature is that the export of HSSW is mainly concentrated on the eastern side of the section (Figs. 11d, g, j and m), which originates from the RIS region." (Lines 492–494 in the revised manuscript).

In addition, S2 is located at the entrance of the Joides Trough, closer to the RIS, which could be significantly affected by the local ice-shelf circulations. Furthermore, Morrison et al. (2020) also presented a consistent northward flow near the east side of S2 around 175°E (Fig. 4B in Morrison et al., 2020).

References:
Morrison, A. K., McC. Hogg, A., England, M. H., and Spence, P.: Warm Circumpolar Deep Water transport toward Antarctica driven by local dense water export in canyons, 6, 1–10, https://doi.org/10.1126/sciadv.aav2516, 2020.

*23. Line 500: Does the barotropic geostrophic current from SSH here be consist with the simulated current field? The significant correlation between the Coriolis term and the pressure gradient term can only indicate geostrophic current, but the barotropic and simulated current need to be evaluated weather it is barotropic.*

Following the reviewers' suggestions, we further calculated the barotropic and baroclinic components of the geostrophic currents to make more solid discussions. A As shown in the revised Fig.14, the barotropic flow over S1 is similar to the ocean currents in the upper layer (revised Figs.15a, d, g and j), which presents a southeastward flow across S1. To further identify the dominant flow component on the defined

transects (S1, S2 and S3), the vertical sections of cross-transect velocity for barotropic and baroclinic geostrophic flow during SYNO1 and MESO were presented by Figs. S8–S11 in the revised supplementary material. The related features for S1 have been added as "In addition, we further examined the barotropic and baroclinic components for this geostrophic flow along S1 (Figs. S8 and S9). The positive (eastward) velocity in the upper layer in the area bounded by 77.1–76.9°S and 76.7–76.5°S (Figs. 11c, f, i and l) is regulated by the barotropic current (Figs. S8a, d, g and j), while the negative (westward) velocity in the deeper layer (Figs. 11c, f, i and l) is related to the baroclinic component resulting from the density differences across S1 (Figs. S9a, d, g and j)." (Lines 549–554 in the revised manuscript). For S3, the updated findings have been added as "Such barotropic currents could be identified on S3 in Fig. S8. Meanwhile, the baroclinic geostrophic flow also plays an important role in HSSW export across S3 (Figs. S9c, f, i and l). Therefore, the northward flow is regulated by both barotropic and baroclinic components. These features for MESO are consistent with that we found for SYNO1 (Figs. S10 and S11)." (Lines 568–572 in the revised manuscript).

*24. Line 504: It may be clearer that plot the geostrophic current or the pressure gradient on Figure 14.*

The barotropic geostrophic currents have been superimposed in this figure as Fig. 14 in the revised version.

*25. Line 508: Do you mean the Ekman transport contributes to the SSH, which lead to the geostrophic current? Or the Ekman transport dominates the current (but it cannot be described as geostrophic current)?*

Sorry for this unclear writing. This sentence aims to illustrate the Ekman transport dominated in the upper layer which contributed to the variation of SSH. The difference of SSH is the main source for the pressure gradient force inducing the surface geostrophic flow. The original sentence "Meanwhile, the relatively strong vertical shear in the upper layer suggests that the Ekman transport could dominate on top of the interior geostrophic current (Figs. 12j and 12t)." has been revised to "Meanwhile, the relatively strong vertical shear in the upper layer suggests that the Ekman transport could dominate on top of the interior geostrophic current (Figs. 12j and 12t), which contributed to the variation of SSH over S1." (Lines 536–538 in the revised version).

*26. Line 556: I think that Morrison et al. (2020) meaned that the intruding mCDW (or CDW) is driven by the outflowing of HSSW but not vice versa, while I agree the mCDW (CDW) can influence the HSSW.*

We agree with the reviewer's comments, and the original sentence mentioning the role of CDW inflow in HSSW flow has been deleted and revised to "Another factor might be the local circulation system like the outflow of basal melting water and the local gyre over these coastal regions. Herraiz-Borreguero et al. (2016) highlight the role of ice shelf water in controlling the HSSW formation rate and its thermohaline properties in East Antarctica. Formation of HSSW could be hindered by the freshwater input from ice shelves (Williams et al., 2016). Meanwhile, the increased freshwater from ice shelf melting could reduce the transport of CDW onto the continental shelf region (Dinniman et al. 2018), which can further affect the formation of dense shelf water." (Lines 589–595 in the revised version).

*27. Line 566: I think that due to the section Mathiot et al. (2012) choosing is influenced by TNB polynya, here claimed the distance from where the HSSW generate (e.g., the distance from S1, S2, S3 to RISP in this paper and the distance from the section in Mathiot et al. (2012) to RISP and TNB polynya) is necessary.*

As the reviewer suggested, the findings of Mathiot et al. (2012) and our study do focus on different time scales, so it does not make much sense to make lag time comparisons. Instead, we tried to address the linkage of this study to our study, and reorganized the statements as "Mathiot et al. (2012) documented a 6-month time lag between the HSSW formation in polynyas (TNB and RISP) and the HSSW transport

across the topographic sills in the Ross Sea, i.e., the maximum HSSW transport occurred during summer (February/March) while the maximum of polynya activity occurs in winter (August/September). The defined sections across the Drygalski Trough and the Joides Trough in their study were located around 74°S, which is about 330 km further north than the sections we selected. This study provided a baseline for us to estimate the timescale for the cyclone-induced sea ice and HSSW change to influence bottom water properties at the slope." (Lines 600–608 in the revised version).

*Reference*
*Morrison, A. K., McC. Hogg, A., England, M. H., & Spence, P. (2020). Warm Circumpolar Deep Water transport toward Antarctica driven by local dense water export in canyons. Science Advances, 6(18), 1–10. https://doi.org/10.1126/sciadv.aav2516*

*Wang, Q., Danilov, S., Hellmer, H., Sidorenko, D., Schroter, J., & Jung, T. (2013). Enhanced cross-shelf exchange by tides in the western Ross Sea. Geophysical Research Letters, 40(21), 5735–5739. https://doi.org/10.1002/2013gl058207*

---

## Author Comment (AC2)

**Response to Review Comments**

for

**"The response of sea ice and high salinity shelf water in the Ross Ice Shelf Polynya to cyclonic atmosphere circulations"**

Xiaoqiao Wang, Zhaoru Zhang, Michael S. Dinniman, Petteri Uotila, Xichen Li, Meng Zhou

**Note**: Reviewers' comments are in italic font; authors' response comments are in normal font. Revisions in the revised manuscript are highlighted by blue color.

**Reviewer comments:**

**Anonymous Referee #2:**

Using a regional ocean-sea ice-ice shelf model they investigate the role of meso and synoptic scale cyclones in sea ice production, HSSW formation and export from the Ross Ice Shelf polynya. The authors found that the Cyclone caused an increase in the sea ice production rate due to changes in offshore winds and a consequently Enhancement of HSSW formation and export.

While I think that this paper could potentially give an important contribution to the understanding of the processes involved in the dense water formation in the Ross Sea, there are several issues that still need to be addressed before publication.

We thank the reviewer for his/her efforts in reviewing this manuscript and providing useful comments that significantly improved our manuscript.

**Overview** Comments:

1. The main results of the manuscript are based on the HSSW salinity and volume increase in the RISP, in response to the increase of sea ice production due to the strengthening of off-shore winds during cyclone events. While the increase of SIP occurs in the RISP (Figures 5 and 9), the increase of HSSW salinity and volume takes place in the region west of Ross Island (Figures 6 and 10), which is not in the RISP polynya, but in what is called the McMurdo polynya. Moreover, in this region there is no increase in SIP, so how do you explain the increase in Salinity there?

I suggest setting the western limit of the RISP at Ross Island (approximately 169.5° E; see Tamura et al., 2008; Orsi and Wiederwohl 2009; Drucker et al., 2011) and recalculating the HSSW salinity and volume. In this case, I suspect that you will not observe anymore a significant increase in HSSW salinity and volume during the cyclones events.

Moreover, from the TS diagram in Figures 6 and 10 is not easy to see the change in salinity of the RISP except for the end-members (higher salinity values), it would be easier to make salinity time-series of the surface, intermediate and the bottom layer at a different location along the RISP.

Sorry for the confusion in the RISP definition. As the reviewer suggested, we redefined the boundary of RISP to make a better separation between the McMurdo polynya and RISP. The western boundary has been changed close to the Ross Island as shown in Fig. 2a in the revised manuscript. In addition, HSSW is redefined as the water mass with neutral density ( $\gamma^n$ ) above 28.27 kg m-3, salinity (S) > 34.62 and potential temperature ( $\theta$ )

---

## Author Comment (AC4)

Response to Review Comments

for

**"The response of sea ice and high salinity shelf water in the Ross Ice Shelf Polynya to cyclonic atmosphere circulations"**

Xiaoqiao Wang, Zhaoru Zhang, Michael S. Dinniman, Petteri Uotila, Xichen Li, Meng Zhou

**Note**: Reviewers' comments are highlighted by blue color; authors' responses are in black color. Revisions in the revised manuscript are highlighted by blue color.

**Reviewer comments:**

**Anonymous Referee #2:**

Using a regional ocean-sea ice-ice shelf model they investigate the role of meso and synoptic scale cyclones in sea ice production, HSSW formation and export from the Ross Ice Shelf polynya. The authors found that the Cyclone caused an increase in the sea ice production rate due to changes in offshore winds and a consequently Enhancement of HSSW formation and export.

While I think that this paper could potentially give an important contribution to the understanding of the processes involved in the dense water formation in the Ross Sea, there are several issues that still need to be addressed before publication.

We thank the reviewer for his/her efforts in reviewing this manuscript and providing useful comments that significantly improved our manuscript.

Overview Comments:

1. The main results of the manuscript are based on the HSSW salinity and volume increase in the RISP, in response to the increase of sea ice production due to the strengthening of off-shore winds during cyclone events. While the increase of SIP occurs in the RISP (Figures 5 and 9), the increase of HSSW salinity and volume takes place in the region west of Ross Island (Figures 6 and 10), which is not in the RISP polynya, but in what is called the McMurdo polynya. Moreover, in this region there is no increase in SIP, so how do you explain the increase in Salinity there?
I suggest setting the western limit of the RISP at Ross Island (approximately 169.5° E; see Tamura et al., 2008; Orsi and Wiederwohl 2009; Drucker et al., 2011) and recalculating the HSSW salinity and volume. In this case, I suspect that you will not observe anymore a significant increase in HSSW salinity and volume during the cyclones events.
Moreover, from the TS diagram in Figures 6 and 10 is not easy to see the change in salinity of the RISP except for the end-members (higher salinity values), it would be easier to make salinity time-series of the surface, intermediate and the bottom layer at a different location along the RISP.

Sorry for the confusion in the RISP definition. As the reviewer suggested, we redefined the boundary of RISP to make a better separation between the McMurdo polynya and RISP. The western boundary has been changed close to the Ross Island as shown in Fig. 2a in the revised manuscript. In addition, HSSW is redefined as the water mass with neutral density ($\gamma^n$) above 28.27 kg m$^{-3}$, salinity (S) $>$ 34.62 and potential temperature ($\theta$) $<$ -1.85° C (Orsi and Wiederwohl, 2009; Castagno et al., 2019) following

Comment #7. All of the related figures and statements have been updated based on this newly defined RISP and HSSW.

As shown in the updated Fig. 6a, HSSW was still mainly formed in the western section of RISP (167°E–176°E) where the increase in SIP can be observed in revised Figs. 5 and 9. In addition, we reproduced the T-S diagrams for selected cyclone events (SYNO1, SYNO2 and MESO) based on the redefined RISP and HSSW in Figs. R1, R2 and R3, which also reveal an increase of HSSW formation in the RISP region (around 167–174°E). The reason why HSSW accumulated in the western RISP may be related to the continuous westward flow along the coastline (at approximately 78°S, 175°E–165°W), which can be observed by Fig. 15 and Fig. S6 in the revised manuscript (particularly prominent in Figs. S6d, e, g, h, f and k). Following the reviewer's suggestion, the T-S diagrams have been modified to the time series of HSSW salinity (Figs. 6, 10 and S3 in the revised version), which still presented similar features to previous T-S diagrams. The relevant statements have been updated based on these time series plots.

References:
Orsi, A. H. and Wiederwohl, C. L.: A recount of Ross Sea waters, Deep. Res. Part II Top. Stud. Oceanogr., 56, 778–795, https://doi.org/10.1016/j.dsr2.2008.10.033, 2009.
Castagno, P., Capozzi, V., DiTullio, G. R., Falco, P., Fusco, G., Rintoul, S. R., Spezie, G., and Budillon, G.: Rebound of shelf water salinity in the Ross Sea, 10, 1–6, https://doi.org/10.1038/s41467-019-13083-8, 2019.

[Figure]

**Fig. R1** (a–l) Temperature–salinity (T–S) diagrams for the RISP region shown in revised Fig. 2a during the SYNO1 event from July 13 to July 19 of 2005. The T–S dots are color-coded with longitude. The black isoline denotes the neutral density contour of 28.27 kg m$^{-3}$.

[Figure]

**Fig. R2** (a–n) Same as Fig. R1 but for the SYNO2 event from September 18 to September 24 of 2014.

[Figure]

**Fig. R3** (a–n) Same as Fig. R1 but for the MESO event from June 21 to June 24 of 2005.

2. One of the main focuses of this work is to estimate the export of HSSW from RISP. If you considered the RISP from 163° to 187° E, why do you draw a meridional transect (S1) in the middle of the polynya? Where the water in S1 is exported from?

Thanks for pointing this out. As mentioned in our reply to Comment #1, the extent of RISP has been revised. For the updated RISP, the meridional transect S1 is located outside the RISP, which makes the calculation of HSSW export reasonable.

3. Because the modelled ocean currents data are crucial to the paper discussion, it would be appropriate to validate those data with in-situ observations. You could use in-situ mooring data in a few areas of the Ross Sea. In the Ross Sea, mooring observations are available from different National programmes (USA, NZ and Italy).

Thanks for this suggestion. It is difficult to conduct point-to-point comparisons, so we have to look at mean pictures of the transport. In Dinniman et al. (2018) with the same model, we mentioned that the pathways were accurate and did look at one mean CDW transport estimate along the western slope of Pennell Bank: "The Ross Sea circulation model accurately simulates the locations of the CDW intrusions [e.g., Fig. 7 in Dinniman et al. (2011); see Fig. S2 in the supplemental material]. Observation-based estimates of total CDW transport onto the Ross Sea continental shelf are limited to a few locations and short periods. The simulated MCDW transport along the western slope of Pennell Bank over a 2-week period in the summer of 2011 [0.22 ± 0.03 Sv (1 Sv ≡ 106 $m^3$ $s^{-1}$); McGillicuddy et al. 2015, see their supplemental material] matched observations made at this location over the same period (0.24 Sv; Kohut et al. 2013), suggesting that the volume input is realistically captured in the simulations.".

References:
Dinniman, M. S., Klinck, J. M., and Smith, W. O.: A model study of Circumpolar Deep Water on the West Antarctic Peninsula and Ross Sea continental shelves, Deep. Res. Part II Top. Stud. Oceanogr., 58, 1508–1523, https://doi.org/10.1016/j.dsr2.2010.11.013, 2011.
Dinniman, M. S., Klinck, J. M., Hofmann, E. E., and Smith, W. O.: Effects of projected changes in wind, atmospheric temperature, and freshwater inflow on the Ross Sea, J. Clim., 31, 1619–1635, https://doi.org/10.1175/JCLI-D-17-0351.1, 2018.
Kohut, J., E. Hunter, and B. Huber, 2013: Small-scale variability of the cross-shelf flow over the outer shelf of the Ross Sea. J. Geophys. Res. Oceans, 118, 1863–1876, https://doi.org/ 10.1002/jgrc.20090.
McGillicuddy, D. J., and Coauthors, 2015: Iron supply and demand in an Antarctic shelf ecosystem. Geophys. Res. Lett., 42, 8088– 8097, https://doi.org/10.1002/2015GL065727.

4. The proposed mechanisms in paragraphs 3.4 and 3.5 are not convincing (in other words, too speculative). I think that the discussions need solid improvements. See below.

Following the reviewer's suggestions, much more calculations and discussions have been conducted to improve the credibility of our study. Please see more detailed information in Comments #17–22.

5. Many works have shown the direct role of the winds on the DSW formation, and because the cyclone influences the local wind dynamics, it is obvious that changes in the position and scale of a cyclone may have slightly different effects on the dense water formation. A more interesting work would be to statistical analysis of the cumulative effect of cyclones on the HSSW formation during the winter season and on the HSSW salinity trends and interannual variability.

It is true that many previous studies have already revealed the effects of winds on water mass formation processes including HSSW and AABW (for instance Mathiot et al., 2010; Barthélemy et al., 2012). The majority of these studies are focused on the seasonal scale or longer time scales. In reality, there are more high-frequency strong wind events (i.e., cyclones) occurring in the Ross Sea and East Antarctica (Uotila

et al., 2011; Turner et al., 2009; Chenoli et al., 2015), and the main purpose of this study is to elucidate the impacts of typical cyclone events on sea ice and HSSW in RISP on a shorter time scale (i.e., synoptic-scale influences) by quantifying the variations and response time for different variables. In our previous study targeted on the Prydz Bay and Shackleton polynyas (Wang et al., 2021), the seasonal effects of strong wind events related to cyclones on HSSW formation have been identified, showing that the duration of strong wind events over the winter season could significantly affect the HSSW formation in the deep ocean. Such investigations are also important for the Ross Sea, but considering the length of the current manuscript (particularly the number of figures), we decided to put such analysis in a future work, and we thank the review for this suggestion.

References:
Barthélemy, A., Goosse, H., Mathiot, P., and Fichefet, T.: Inclusion of a katabatic wind correction in a coarse-resolution global coupled climate model, Ocean Model., 48, 45–54, https://doi.org/10.1016/j.ocemod.2012.03.002, 2012.
Chenoli, S. N., Turner, J., and Samah, A. A.: A strong wind event on the ross ice shelf, antarctica: A case study of scale interactions, Mon. Weather Rev., 143, 4163–4180, https://doi.org/10.1175/MWR-D-15-0002.1, 2015.
Mathiot, P., Barnier, B., Gallée, H., Molines, J. M., Sommer, J. Le, Juza, M., and Penduff, T.: Introducing katabatic winds in global ERA40 fields to simulate their impacts on the Southern Ocean and sea-ice, Ocean Model., 35, 146–160, https://doi.org/10.1016/j.ocemod.2010.07.001, 2010.
Turner, J., Chenoli, S. N., Abu Samah, A., Marshall, G., Phillips, T., and Orr, A.: Strong wind events in the Antarctic, J. Geophys. Res. Atmos., 114, https://doi.org/10.1029/2008JD011642, 2009.
Uotila, P., Vihma, T., Pezza, A. B., Simmonds, I., Keay, K., and Lynch, A. H.: Relationships between Antarctic cyclones and surface conditions as derived from high-resolution numerical weather prediction data, J. Geophys. Res. Atmos., 116, 1–14, https://doi.org/10.1029/2010JD015358, 2011.
Wang, X., Zhang, Z., Wang, X., Vihma, T., Zhou, M., Yu, L., Uotila, P., and Sein, D. V.: Impacts of strong wind events on sea ice and water mass properties in Antarctic coastal polynyas, Clim. Dyn., 57, 3505–3528, https://doi.org/10.1007/s00382-021-05878-7, 2021.

Specific Comments:

6. Line 28-30: in this paper, you have chosen to represent positive along shelf velocity with a westward current, but in general in Oceanography a positive zonal component is considered to be positive in the eastward direction, so it is a bit confusing. Therefore, here I suggest explicit the direction of the correlation: transport direction (eastward or westward) with the wind direction (northward or southward).

The definition of along-shelf velocity sign in the original manuscript is indeed not common in the oceanography community, though we wanted to define the direction of HSSW export toward the slope (westward) as positive. As the reviewer suggested, we changed the definition and now the eastward direction is defined positive in the revised manuscript. All related texts and figures (revised Figs. 7 and 11) have been updated using this new definition.

7. Line 190-191: In order to define and identify HSSW in the Ross Sea I suggest using the definition proposed by Orsi and Wiederwohl (2009) that is more commonly used by the Ross Sea community. This definition uses both traditional thermohaline parameters (potential temperature and salinity S) and neutral density.

The definition of HSSW has been revised according to the reviewer's suggestion. Please see our detailed response to Comment #1.

8. Line 216-221: The sentence "three-dimensional along-shelf and cross-shelf momentum equations are..., where $\frac{\partial u}{\partial t}$ and $\frac{\partial v}{\partial t}$ are the alongshore and cross-shore components of velocity," is a bit confusing. it is not clear if the currents are along-shelf (parallel to the shelf-break?) or along-shore (along the Ross Ice shelf?). I suggest to use along and across the Ice shelf.

The original sentence has been revised to "The three-dimensional along-ice-shelf and across-ice-shelf (defined by local acceleration terms) momentum equations are…where $u$ and $v$ are the along-ice-shelf and across-ice-shelf components of velocity…" to make the direction clear (Lines 224–229 in the revised version). Furthermore, the model girds have been shown in Fig. 1b to illustrate the along- and across-ice-shelf directions.

9. Lines 236-237: looking to figure 3a I can see the lowest density in the western ross sea (dark blue), lower than the eastern Ross Sea (light blue) and the highest density outside the Ross Sea, west of Cape Adare.

Sorry for the inaccurate expression. The original sentence "The high track density of synoptic-scale cyclones extends to the continental shelf and coastal regions in the western Ross Sea (Fig. 3a)." has been revised to "The high track density of synoptic-scale cyclones extends to the continental slope regions of the western Ross Sea (at around 65°S, Fig. 3a)" (Lines 246–247 in the revised version).

10. Lines 237-238: you should change the wording to explain the figure better. I see a higher density at 180° outside the continental shelf and not close to the RIS.

The original sentence "For mesoscale cyclones, a large number of track densities appear at the center of the Ross Sea (at about 180° meridional) in front of the RIS central region (Fig. 3b), which…" has been revised to "For mesoscale cyclones, on the Ross Sea continental shelf a large number of track densities appear in front of the RIS central region (near the ~180° meridian, Fig. 3b), which…" to make a clearer statement (Lines 247–248 in the revised version).

11. Line 294: Salinity is overestimated (See Orsi and Wiederwohl 2009; Jacobs et al., 2022). The region 163°E–164°E is not in the RISP (Tamura et al., 2008; Orsi and Wiederwohl 2009; Drucker et al., 2011).

As mentioned in our response to Comment #1, the western boundary of RISP has been revised. The updated salinity values are more reasonable and lower than previous ones (Figs. 6, 10 and S3 in the revised version).

12. Lines 294-296: Here you suggest that the increase in salinity occurs in the region 163°E–164°E, when is observed an increase in SIP. From figure 5, I can see the increase in SIP in the region east of Ross Island (169.5), how the increase in salinity in the region 163°-164° E (that is outside the RISP) is explained by the increase in SIP west of 169.5 at about the same time (The distance between these 2 regions is not less than 150 Km)?

The initial idea was that the SIP change in the region east of the Ross Island can affect the HSSW on the west by the westward coastal current along the RIS, which could be seen in the upper layers in Fig. 15 and Fig. S7 (particularly eminent in Figs. S7d, e, g, h, f and k). Meanwhile, we apologize for the misunderstanding here. What we were trying to demonstrate was that SIP can regulate changes in the salinity of HSSW on the western RISP, but did not emphasize the instantaneous correlation between these two variables. Instead, the persistence of the increase in HSSW salinity indicates that there is a lagging or cumulative effect of the HSSW response to SIP. Anyway, in summary, the T-S diagrams have been modified into time series plots to facilitate a better interpretation of the changes in HSSW salinity (Figs. 6, 10 and S3 in the revised version).

13. Lines 319-321: I think this expression is misleading. In this paper, the positive velocity is considered westward, whilst usually, the positive zonal velocity is considered to be eastward.

Please see our response to Comment #6. The related sentences have been revised to "Meanwhile, there was notable change in the cross-transect current velocity: the positive (eastward) velocity in the upper layer between 76.7°S and 76.5°S decreased significantly, while the negative (westward) velocity in the bottom layer increased (Figs. 8f to i and l). Both features could lead to reduced eastward exports when the wind speed decreased." (Lines 345–348 in the revised version).

14. Line 368: Also in SYNO2 the salinity increases mostly west of RI, and not in the RISP.

Please see our response to Comment #1.

15. Lines 369-371: I do not think that the HSSW salinity decreased in relation to the decrease in SIP on the eastern side of the RISP. During stage I of SYNO2 (Figure S3) the salinity increases mostly west of Ross island, therefore most probably, HSSW salinity is not affected by the SIP decrease in the eastern Ross Sea. Moreover, In the Eastern RIS, the salinity is much lower compared to the western Ross Sea and there is no HSSW production (Orsi and Wiederwohl 2009), therefore is not clear how the SIP reduction in the eastern RISP helps to decrease the salinity of the HSSW.

Based on the updated results in the revised version, the related statements in this part have been modified to "For the water mass response, HSSW volume variability in the RISP increased significantly until 00:00 on 21 September and then remained positive values for at least 60 hours (Fig. S3a). The salinity of newly formed HSSW increased to 34.84 psu at 00:00 of September 21 after Stage I and II of the SYNO2 cyclone (Figs. S3b). Afterwards, the volume and salinity of HSSW kept increasing for 36 hours when the coastal SIP was already decreasing (Figs. S2 and S3)." (Lines 386–390 in the revised version).

16. Lines 465-466: you haven't mentioned before that at S1 in Syno1 there is a 12 hours lag between HSSW export and the wind.

Sorry for this typo. S1 should be changed to S3 based on the previous results. The original sentence has been revised to "As there are lag correlations between wind speed and current velocity both for S1 and S3, such a relationship can explain why the HSSW export also exhibited lag responses to the wind speed." according to the updated results (Lines 483–485 in the revised version).

17. Lines 489-490: I suggest showing the figure in the supplementary material.

Following the reviewer's suggestion, the momentum analysis results for SYNO1 are shown in Figs. S4 and S5 in the revised supplementary material, and these figures are also mentioned in the text as "For SYNO1 (Figs. S4 and S5), the momentum balance presents similar results to those of MESO." (Lines 514–515 in the revised version).

18. Line 491: "are displayed in the cross-shelf (Fig. 12) and along-shelf (Fig. 13) directions respectively." Is along-shelf in figure 12 (Transect S1) and cross-shelf in Figure 13 (transect S3)?

No, the momentum equation used for the meridional transect S1 is for the across-ice-shelf component ($\frac{\partial v}{\partial t} = -fu...$, which defined by local acceleration terms). For the zonal transect S3, the across-transect velocity is regulated by the along-ice-shelf equation ($\frac{\partial u}{\partial t} = fv...$). To make a clearer statement, related sentences have been added in the revised version as we mentioned in Comment #8.

19. Lines 504-507: In Figure S4, I can't see the intensified westward Ekman transport in the region 74°-76.5° S and 163°-167° E. Moreover, Why the increase in the off-shore winds intensify the westward Ekman transport only in that region of the Ross Sea? Please explain.
In addition, in case there is an increase in the pressure gradient between 74°-76.5° region and the RIS, why do we observe the increase in the eastward transport only in a part of transect S1 and not in the whole transect?
Furthermore, the geostrophic flow due to a tilt in the sea surface slope should be mostly barotropic. In figure 7 (and 11) does not look like the transport is barotropic.
You should also consider that the pressure gradients depend both on the sea surface slope and the density gradient between the RIS and the outer shelf. During cyclones, the increase in SIP and salinity close to the RIS should enhance the geostrophic flow due to the salinity differences.

The region bounded by 74°–76.5°S and 163°–167°E has been highlighted for better revealing of the westward Ekman transport (marked by the blue boxes in Figs.15d and 15j in the revised manuscript). An explanation as to why enhanced westward Ekman transport was only observed in this region is actually given in the original version, but may not be clear enough. The first reason is that during the MESO event, only the west section of RISP (approximately west of 180°) was dominated by offshore winds (updated Fig. 8), which leads to the westward Ekman transport. Moreover, a southeastward flow persisted close to the Ross Island, which was captured over the entire depth (Fig. 15 in the revised version). The water transport carried by this southeastward flow could result in a consistent lower sea surface elevation in this region near S1 (revised Fig. 14), then inducing a continuous pressure gradient force that plays a dominant role on the local currents. Therefore, the Ekman transport would not work in this region. The detailed information about "the southeastward flow" has been addressed in Comment #20. So to conclude, the intensified Ekman transport is only observed in the northwest region of the Ross Sea.

To give a better interpretation of this part. The sentence "These features suggest that the increased offshore winds induced intensified westward Ekman transports in the upper layer over 74–76.5°S and 163–173°E just north of S1 (Fig. S4), eventually resulting in the higher sea surface elevation in this region (Figs. 14b and 14d). Meanwhile, the relatively strong vertical shear in the upper layer suggests that the Ekman transport could dominate on top of the interior geostrophic current (Figs. 12j and 12t). However, near the RIS over 163–173°E where offshore winds also prevailed, the increase in surface elevation could barely be detected." has been revised to "These features suggest that the increased offshore winds induced intensified westward Ekman transports in the upper layer in the area bounded by 74–76.5°S and 163–176°E just north of S1 (marked by the blue box in Figs. 15d and 15j), eventually resulting in the higher sea surface in this region (Figs. 14b and 14d). Meanwhile, the relatively strong vertical shear in the upper layer suggests that the Ekman transport could dominate on top of the interior geostrophic current (Figs. 13j and 13t), which contributed to the variation of SSH over S1. However, near the RIS between 163°E and 176°E where offshore winds also prevailed, the increase in surface elevation (i.e., the enhanced westward Ekman transport) could barely be detected (Fig. 15)." (Lines 534–542 in the revised version).

Following the reviewers' suggestions, we further calculated the barotropic and baroclinic components of the geostrophic currents to make more solid discussions. A As shown in the revised Fig.14, the barotropic flow over S1 is similar to the ocean currents in the upper layer (revised Figs.15a, d, g and j), which presents a southeastward flow across S1. To further identify the dominant flow component on the defined transects (S1, S2 and S3), the vertical sections of cross-transect velocity for barotropic and baroclinic geostrophic flow during SYNO1 and MESO were presented by Figs. S8–S11 in the revised supplementary material. The related features for S1 have been added as "In addition, we further examined the barotropic and baroclinic components for this geostrophic flow along S1 (Figs. S8 and S9). The positive (eastward) velocity in the upper layer in the area bounded by 77.1–76.9°S and 76.7–76.5°S (Figs. 11c, f, i and l) is regulated by the barotropic current (Figs. S8a, d, g and j), while the negative (westward) velocity in the deeper layer (Figs. 11c, f, i and l) is related to the baroclinic component resulting from the density differences across S1 (Figs. S9a, d, g and j)." (Lines 551–556 in the revised manuscript). For S3,

the updated findings have been added as "Such barotropic currents could be identified on S3 in Fig. S8. Meanwhile, the baroclinic geostrophic flow also plays an important role in HSSW export across S3 (Figs. S9c, f, i and l). Therefore, the northward flow is regulated by both barotropic and baroclinic components. These features for MESO are consistent with that we found for SYNO1 (Figs. S10 and S11)." (Lines 570–574 in the revised manuscript).

20. Lines 511-516: In Figure S4, I can barely see the eastward flow in the upper layer. Looking at the figure all I can see is a very chaotic circulation with currents directed in the opposite direction at close nodes, especially close to the RIS and Ross Island. Moreover, along the RIS, a strong coastal current has been observed that flows westward and not eastward (Pillsbury & Jacobs 1985; Jacobs & Giulivi 1998; Smith et al., 2012; Porter et al., 2019).
Could it be useful to show a figure with the mean surface circulation across S1? Furthermore, I do not think that in the upper layer (50 m) the HSSW can flow underneath the RIS. The RIS thickness at the front is no less than 150-200 m thick, well below the depth of 50 m.
Moreover, both Budillon et al. (2003) and Jendersie et al. (2018) show an inflow of HSSW underneath the RIS in the deep layer and not in the upper 50 m layer as observed here.

We are sorry that we found a mistake over the RIS region when producing the original Fig. S4, and the current patterns outside the ice shelf region were correct. In the revised figures, there are no flows at 50 m over the RIS region (see revised Fig. 15 and R4), which is reasonable now. Meanwhile, the spatial pattern of averaged circulation for the surface, intermediate, and bottom layers has been presented in Fig. R4 as the reviewer suggested. By comparing the features between Fig. R4 and original Fig. S4 (now revised as Fig. 15 in the main text), there are no big differences and still some chaotic currents over S1. Therefore, to more clearly demonstrate the eastward flow mentioned in the main text, we have marked the relevant area with yellow boxes near S1 (updated Fig. 15). By examining the current patterns within the boxed area over S1, there was a southeastward flow that could be observed at depths of 300 and 500 m and can further flow underneath the RIS reach around 78°S. The components of this southeastward flow in different directions can be clearly shown in the revised Fig. S6 and Fig. S7 respectively. So generally, this flow is consistent with the HSSW inflow mentioned by Budillon et al. (2003) and Jendersie et al. (2018). In addition, the strong westward flow along the RIS mentioned by the reviewer could also be found in Fig. S6 (indicated in blue along coastlines).

The related statement "After examining the horizontal pattern of currents over the Ross Sea, we found a current flowing from west to east across S1 located north of the Ross Island, which further flowed southward below the RIS (Fig. S4) in the upper layer." has been modified to "After examining the horizontal pattern of currents over the Ross Sea, we found a southeastward flow across S1 located north of the Ross Island (within the yellow boxed area near S1 in Fig. 15), which can also be detected in Fig. 14, suggesting that it can be regarded as the barotropic flow resulting from sea surface change. Furthermore, this southeastward current further flowed southward below the RIS in the deeper layer (Figs. 15b, c, e, f, h, i, k and l), and the zonal and meridional components of this southeast flow can be observed more clearly in Fig. S6 and Fig. S7 respectively." to better explain this phenomenon (Lines 542–547 in the revised version).

[Figure]

**Figure R4**. Spatial distributions of depth-averaged ocean currents over (a, d, g and j) 0–50 m, (b, e, h and k) 50–400 m and (c, f, i and l) 400–800 m at four selected time points (06:00 am of June 21, 12:00 am of June 22, 12:00 of June 23 and 06:00 of June 24). The red lines are the S1, S2, and S3 sections defined in Fig. 1b

21. Lines 545-552: I do not agree with the sentence "The HSSW formation in the RISP demonstrated a near-instantaneous response to the wind change during the synoptic- and meso-scale cyclone events". Here, the increase of HSSW salinity and volume is observed west of Ross Island and not in the RISP polynya where the SIP increases during the Cyclones. This is clear from Figures 6 and 10. Looking at the TS diagram confirms that the HSSW salinity increase occurs west of Ross Island, where there is no increase in SIP. Therefore, the discussion below is pointless.

Based on the redefined HSSW and RISP, the updated results suggest that the HSSW is mainly formed within the RISP region (around 167°E–174°E). Please see more detailed information in our response to Comment #1. Meanwhile, the HSSW volume also showed instantaneous increase when the wind was increasing (Figs. 6b in the revised version).

22. Lines 553-568: Following Morrison et al., 2020 the CDW intrusion and HSSW outflow are correlated, but close to the shelf break and not close to the RIS. In addition, it is the HSSW outflow that modulates the CDW intrusion and not the contrary as stated here. While it is true that the CDW may affect the DSW formation rate in the Ross Sea, the time scales are very different. The response of the RISP and therefore of the HSSW formation to a cyclone (strong winds) is much shorter than the response of the dense water formation to the CDW intrusion onto the shelf. Moreover, the spatial scales are completely different: the continental shelf in the Amundsen Sea is smaller and the ice shelf is much closer to the shelf break than in the Ross Sea. In addition, the continental shelf water properties in the Amundsen Sea are completely different from the Ross Sea, in the Amundsen Sea, there is no DSW formation.

Finally, I don't understand why you compare the time lag registered in your study (related to the local ocean dynamics response to the northward wind increase during a cyclone, with Mathiot et al., (2012) work that looks at the lag between the cumulative HSSW production during summer and export from the Ross Sea. These are processes happening at different time scales.

We agree with the reviewer's comments, and the original sentence mentioning the role of CDW inflow in HSSW flow has been deleted and revised to "Another factor might be the local circulation system like the outflow of basal melting water and the local gyre over these coastal regions. Herraiz-Borreguero et al. (2016) highlight the role of ice shelf water in controlling the HSSW formation rate and its thermohaline properties in East Antarctica. Formation of HSSW could be hindered by the freshwater input from ice shelves (Williams et al., 2016). Meanwhile, the increased freshwater from ice shelf melting could reduce the transport of CDW onto the continental shelf region (Dinniman et al. 2018), which can further affect the formation of dense shelf water." (Lines 591–597 in the revised version).

As the reviewer suggested, the findings of Mathiot et al. (2012) and our study do focus on different time scales, so it does not make much sense to make lag time comparisons. Instead, we tried to address the linkage of this study to our study, and reorganized the statements as "Mathiot et al. (2012) documented a 6-month time lag between the HSSW formation in polynyas (TNB and RISP) and the HSSW transport across the topographic sills in the Ross Sea, i.e., the maximum HSSW transport occurred during summer (February/March) while the maximum of polynya activity occurs in winter (August/September). The defined sections across the Drygalski Trough and the Joides Trough in their study were located around 74°S, which is about 330 km further north than the sections we selected. This study provided a baseline for us to estimate the timescale for the cyclone-induced sea ice and HSSW change to influence bottom water properties at the slope." (Lines 602–608 in the revised version).

23. Figure 3: I suggest adding the trajectory of the cyclones in the figures.
Figure S4: Because this figure is important for the discussion, I suggest including it in the main text. I also suggest highlighting the region 74°-76.5° S and 163°-167° E.

The cyclone tracks of three typical cases (SYNO1, SYNO2 and MESO) are added to Fig. 3 as the reviewer suggested, and descriptions of the tracks and cyclone evolutions are added in the revised version (Lines 251–263).

Figure S4 in the original supplementary material has been added to the revised main text as Fig. 15. In addition, the region 74°–76.5°S and 163°–167°E has been highlighted as the reviewer suggested. Please see the detailed response in Comment #19 and #20.

References:
Drucker, R., Martin, S., & Kwok, R. (2011). Sea ice production and export from coastal polynyas in the Weddell and Ross Seas. Geophysical Research Letters, 38(17), 4–7. https://doi.org/10.1029/2011GL048668

Jacobs, S. S., & Giulivi, C. F. (1998). Thermohaline Data and Ocean Circulation on the Ross Sea Continental Shelf. https://doi.org/10.1007/978-88-470-2250-8

Jacobs, S. S., Giulivi, C. F., & Dutrieux, P. (2022). Persistent Ross Sea Freshening From Imbalance West Antarctic Ice Shelf Melting. Journal of Geophysical Research: Oceans, 127(3), 1–19. https://doi.org/10.1029/2021jc017808

Orsi, A. H., & Wiederwohl, C. L. (2009). A recount of Ross Sea waters. Deep Sea Research Part II: Topical Studies in Oceanography, 56(13–14), 778–795. https://doi.org/10.1016/j.dsr2.2008.10.033

Pillsbury, R. D., & Jacobs, S. S. (1985). Preliminary observations from long-term current meter moorings near the Ross Ice Shelf, Antarctica. In Oceanology of the Antarctic Continental Shelf, Antarctic Research Series (Vol. 43, pp. 87–107). Washington, D.C: American Geophysical Union. http://doi.org/10.1029/AR043p0087.

Porter, D. F., Springer, S. R., Padman, L., Fricker, H. A., Tinto, K. J., Riser, S. C., et al. (2019). Evolution of the seasonal surface mixed layer of the Ross Sea, Antarctica, observed with autonomous profiling floats. Journal of Geophysical Research: Oceans. https://doi.org/10.1029/2018JC014683

Smith JR, W. O., Sedwick, P. N., Arrigo, K. R., Ainley, D. G., & Orsi, A. H. (2012). The Ross Sea in a sea of Change. Oceanography, 25(3), 90–103.

---

## Author Response (AR2)

**Response to 2nd Review Comments**

for

**"The response of sea ice and high salinity shelf water in the Ross Ice Shelf Polynya to cyclonic atmosphere circulations"**

Xiaoqiao Wang, Zhaoru Zhang, Michael S. Dinniman, Petteri Uotila, Xichen Li, Meng Zhou

**Note**: Reviewers' comments are highlighted by blue color; authors' responses are in black color. Revisions in the revised manuscript are highlighted by blue color.

**Reviewer comments:**

**Anonymous Referee #1:**

The authors have addressed most of my comments very nicely. In general, I am satisfied with the authors' efforts. Her are only a few minor comments.

We thank the reviewer for his/her efforts in reviewing this manuscript and providing useful and helpful comments that further improved our manuscript.

1. With Figures R1 and R2, at ~177W, there is a large region with thick sea ice and low sea ice concentration, which extends to the north. Without the thick ice covered, which can inhibit the ocean from losing heat to the atmosphere, there should be an active heat exchange and sea ice production. However, it is not reflected in Figure 5. Moreover, if we identify the polynya by ice thickness (e.g., <20 cm, Nihashi et al., 2017), or sea ice concentration (e.g., < 0.7, Ding et al., 2020), I think this region will be considered as a part of RSP. But when the polynya is identified by SIP, the extent of the polynya obviously does not include this region. So the inconsistency between the SIP and ice thickness and sea ice concentration may affect the calculation of the polynya area. So, I think it's necessary to explain why low SIP occurs in the thin ice regions.

Reference

Ding, Y., Cheng, X., Li, X., Shokr, M., Yuan, J., Yang, Q., & Hui, F. (2020). Specific relationship between the surface air temperature and the area of the Terra Nova Bay polynya, Antarctica. Advances in Atmospheric Sciences, 37(5), 532–544. https://doi.org/10.1007/s00376-020-9146-2 Nihashi, S., Ohshima, K. I., & Tamura, T. (2017). Sea-Ice Production in Antarctic Coastal Polynyas Estimated From AMSR2 Data and Its Validation Using AMSR-E and SSM/I-SSMIS Data. Ieee Journal of Selected Topics in Applied Earth Observations and Remote Sensing, 10(9), 3912–3922. https://doi.org/10.1109/Jstars.2017.2731995

Figures R1 and R2 are the snapshots of sea ice thickness and sea ice concentration patterns during the first synoptic-scale cyclone event, which was affected significantly by the synoptic process. However, the identification of polynyas is usually determined by the average value from April to October (Ding et al., 2020) or June to September (Kern., 2009). By examining the spatial distribution of multiyear averaged sea ice concentration and sea ice thickness in our study, it actually shows consistent features with sea ice production, i.e., thick ice and low production.

Reference:

Kern, S., 2009: Wintertime Antarctic coastal polynya area: 1992–2008. Geophys. Res. Lett., 36, L14501, https://doi.org/10. 1029/2009GL038062.

2. Line 394–395: considering the large P-value here, the correlation between the meridional wind speed and the HSSW export 12-hours later could be described as "significant but also weak".

Following the reviewer's suggestion, the original sentence "For S3 on the other hand, there was a positive significant correlation between the meridional wind speed and the HSSW export 12-hours later (R=0.53, P=0.042)." has been revised to "For S3 on the other hand, there was a significant but weak positive correlation between the meridional wind speed and the HSSW export 12-hours later (R=0.53, P=0.042)." (Lines 402–404 in the revised version).

3. The red lines of sections and yellow boxes in Figure 15 are difficult to see. Maybe its color should be changed.

The red lines and yellow boxes have been modified to more visible colors in the revised Figure 15.

**Anonymous Referee #3:**

Here are some special comments (line number refer to the version with tracked changes):

1) Line 148 "AMSR-E SIP", could you give some assessments on the potential errors for this product? Is it sensitive to the synoptic process, e.g., the increased cloud cover and humidity during the cyclone?

As mentioned in the original manuscript, the derived AMSR-E SIP is estimated based on a simplified heat flux calculation that ignores the oceanic heat flux for sea-ice freezing/melting process, and therefore the contribution of oceanic heat flux could not be presented by this product. Meanwhile, the AMSR-E SIP product does not discriminate the active-frazil area for Antarctic coastal polynyas, which could result in an underestimation of sea ice production for frazil-dominant polynyas (Nakata et al., 2021). On the AMSR-E SIP website (http://www.lowtem.hokudai.ac.jp/wwwod/polar-seaflux/), the estimated sea ice production dataset (the data used in this study) is available as monthly results, and the effects of synoptic processes on ice production could not be well captured based on the monthly outputs. The increased cloud cover during the cyclones could induce the overestimated ice thickness (Tamura et al., 2007), as cloudiness is used in the heat flux calculation (Nihashi and Ohshima, 2015). The thicker ice thickness suggests smaller heat loss, which finally results in lower ice production based on the AMSR-E dataset. The increased humidity gradients could reduce ice production and sea ice concentration by increasing turbulent fluxes (Taylor et al., 2018). The discussions of potential errors of the AMSR-E SIP production are added as "In addition, the AMSR-E SIP product does not discriminate the active-frazil area for Antarctic coastal polynyas, which could result in an underestimation of sea ice production for frazildominant polynyas (Nakata et al., 2021)." in Lines 182-184 of the revised version.

References:

Nakata, K., Ohshima, K. I., & Nihashi, S. (2021). Mapping of active frazil for Antarctic coastal polynyas, with an estimation of sea-ice production. Geophysical Research Letters, 48, e2020GL091353. https://doi.org/10.1029/2020GL091353

Nihashi, S. and Ohshima, K. I. (2015). Circumpolar Mapping of Antarctic Coastal Polynyas and Landfast Sea Ice: Relationship and Variability, J. Clim., 28, 3650–3670, https://doi.org/10.1175/JCLI-D-14-00369.1.

Tamura, T., Ohshima, K. I., Markus, T., Cavalieri, D. J., Nihashi, S., & Hirasawa, N. (2007). Estimation of Thin Ice Thickness and Detection of Fast Ice from SSM/I Data in the Antarctic Ocean, Journal of Atmospheric and Oceanic Technology, 24(10), 1757-1772.

Taylor, P.C., Hegyi, B.M., Boeke, R.C., Boisvert, L.N. (2018). On the Increasing Importance of Air-Sea Exchanges in a Thawing Arctic: A Review. Atmosphere, 9, 41. https://doi.org/10.3390/atmos9020041.

2) Line 153 "ERA-Interim reanalysis" could you explain why don't you use the updated version of ERA reanalysis data (ERA-5), that has a higher resolution.

The primary reason for using ERA-Interim reanalysis products is that the model was developed early, and long-term simulations were conducted before the ERA5 product came out. In this study, we picked up key periods within the long-term simulation under the influence of synoptic- or mesoscale cyclones, and re-ran the model to generate 6-hourly output (the original output as 5-day-average) to study the impacts of cyclones with short lifetime. For consistency between the atmospheric forcings for the restart fields and model re-run, we used ERA-Interim product for the re-run as well. In addition, we used to examine the sea level pressure (SLP) fields during the passage of cyclones for other Southern Ocean regions for both ERA-Interim and ERA5, such as the Prydz Bay in our previous study, and found the SLP patterns are very similar between the two products, and there are also strong correlations between wind speed from the two datasets. Combining this fact and the consideration for model consistency as mentioned above, we maintained using ERA-Interim as the forcing fields for the model simulation during cyclone events.

3) Line 158 "between wind speed from ERA-Interim and observations": Where did the observation data come from?

The observational data come from the Reference Antarctic Data for Environmental Research (READER) project website (http://legacy.bas.ac.uk/met/ READER), which has been mentioned in the original manuscript "The wind speed data are available at the Reference Antarctic Data for Environmental Research (READER) project website.". To make this clear, this sentence has been revised to "The observed wind speed data are available at the Reference Antarctic Data for Environmental Research (READER) project website (http://legacy.bas.ac.uk/met/READER)." (Lines 154–156 in the revised version).

4) Line 184 "Correlation coefficients between the modeled and observed sea ice concentration": Absolute and relative deviations are also very important.

As the reviewer suggested, we calculated the absolute and relative deviations for the modeled sea ice concentration in 2005 and 2014. The absolute deviations are -0.31 and -0.29 for 2005 and 2014 respectively. The relative deviations for 2005 and 2014 are respectively 36.7% and 34.6%. Generally, our study focuses on the temporal variation of sea ice concentration, i.e., the changes before and after these cyclones, so the correlation coefficients are much more important compared with these deviations. Still in the revised text, we added the information of absolute deviations of modeled ice concentration (Lines 189–190 in the revised version).

5) Line 216 "The cyclones were categorized into two types depending on their horizontal scale": What are the essential differences between these two kinds of cyclones the horizontal size? How to judge these cyclone processes does not have the superimposed contributions from the katabatic wind?

The synoptic-scale weather systems typically have a horizontal length scale in the range of 1000–6000 km and a lifetime of between one day and a week. The mesoscale cyclones are relatively short-lived, sub-synoptic-scale low-pressure systems, and their limited horizontal scale of less than 1000 km and lifecycle of normally under 24 h (King and Turner., 1997). Therefore, the threshold in our study is 1000 km to identify these two types of cyclones, following Uotila et al. (2013). The synoptic-scale depressions form mainly on stronger horizontal temperature gradients (baroclinic zones) in the troposphere and grow

through baroclinic instability. Strong gradients of sea surface temperature are also responsible for the establishment or reinforcement of atmospheric baroclinic zones within which synoptic-scale cyclone can develop. However, many atmospheric thermal gradients near ocean fronts will be relatively shallow features and most of the vortices developing there will be mesocyclones rather than major synoptic systems. Turner and Thomas (1994) also demonstrated that mesocyclones are predominantly an oceanic phenomenon. The mesocyclones are usually the cold air vortices forming to the south of the main polar front and develop in outbreaks of polar air well removed from pre-existing frontal cloud bands. Such information is added in Lines 222–224 of the revised version.

There are two definitions for the katabatic wind. The narrow definition is the air flows generated by the combined effects of gravitational force and pressure gradient (due to radiative cooling of air over glacier and thermal gradient in air above the glacier and above sea) force near the Antarctic coast, which is a local feature. The wide definition is "downslope air flow", which includes the effect of cyclones (Parish and Cassano, 2003). The forcing fields used in our study come from the ERA-Interim reanalysis products which assimilated data from several Antarctic meteorological stations, and should include some information of local katabatic wind. However, the coarse spatial resolution of these products cannot fully represent the small-scale local katabatic wind. If we consider the wide definition of katabatic wind, it is largely affected by the interactions of cyclones with the geography, and the cyclones can be well captured in the reanalysis products.

**References:**

King, J., & Turner, J. (1997). Antarctic Meteorology and Climatology (Cambridge Atmospheric and Space Science Series). Cambridge: Cambridge University Press. doi:10.1017/CBO9780511524967.

Parish, T. and Cassano, J. (2003). The Role of Katabatic Winds on the Antarctic Surface Wind Regime, Mon. Weather Rev., 131, 317–333.

Turner, J. and Thomas, J.P. (1994), Summer-season mesoscale cyclones in the bellingshausen-weddell region of the antarctic and links with the synoptic-scale environment. Int. J. Climatol., 14: 871-894. https://doi.org/10.1002/joc.3370140805.

Uotila, P., Vihma, T., and Tsukernik, M. (2013). Close interactions between the Antarctic cyclone budget and large-scale atmospheric circulation, Geophys. Res. Lett., 40, 3237–3241, https://doi.org/10.1002/grl.50560.

6) Line 268: "indicate complete cyclone trajectories of selected cases": Could you show the time series of the trajectories in the illustration?

Following the reviewer's suggestion, the trajectories in Figure 3 have been modified by adding the start and end positions (indicated by different markers) to present the change in positions over time, The detailed information has been added as "... the pentacles indicate the starting positions and the diamonds present the ending positions." (Lines 276–277 in the revised version).

**7) Line 398: "such as ice shelf circulations": What do you mean the ice shelf circulations here?**

Sorry for the confusion. The ice shelf circulation mainly refers to the circulations of ice shelf basal melting water, so this sentence has been modified to "other factors (such as the circulations of ice shelf basal melting water) could regulate the HSSW exports significantly." (Lines 407–408 in the revised version).

8) "4 Conclusions": Differences of impacts on productions of sea ice and high-salt shelf water between synoptic- and meso-scale cyclones are still not clear. In the conclusions, you should make it more distinct, and highlight it.

Thanks for the reviewer's suggestion, the original statement "When synoptic-scale cyclones prevailed over this region, the entire RISP was dominated by strong offshore winds, which resulted in increased

SIP rates. During the passage of the mesoscale cyclone, SIP increased rapidly over the western side of RISP but decreased over the eastern side of RISP, due to changes in the offshore winds associated with the cyclonic wind field." have been revised to "When synoptic-scale cyclones with spatial size over 1000 km prevailed over this region, the entire RISP was dominated by strong offshore winds, which resulted in increased SIP rates in the entire RISP. While during the passage of mesoscale cyclones with radii less than 1000 km, SIP increased rapidly over the western side of RISP but decreased over the eastern side of RISP, due to changes in the offshore winds associated with the cyclonic wind field.", in order to emphasize the difference impacts of the two cyclones types in ice production (Lines 624–629 in the revised version). Meanwhile, the sentence "The main differences in the response of HSSW formation to the synoptic- and mesoscale cyclones lie in the persistent time of high-salinity signals after the cyclone decayed." has been added in the revised version (Lines 632–633) to clarify the differences in HSSW formation between synoptic- and meso-scale cyclones.